# Physical determinants of vesicle mobility and supply at a central synapse

Jason Seth Rothman[1], Laszlo Kocsis[2], Etienne Herzog[3,4], Zoltan Nusser[2]*, Robin Angus Silver[1]*

[1]Department of Neuroscience, Physiology and Pharmacology, University College London, London, United Kingdom; [2]Laboratory of Cellular Neurophysiology, Institute of Experimental Medicine, Hungarian Academy of Sciences, Budapest, Hungary; [3]Department of Molecular Neurobiology, Max Planck Institute of Experimental Medicine, Göttingen, Germany; [4]Team Synapse in Cognition, Interdisciplinary Institute for Neuroscience, Université de Bordeaux, UMR 5297, F-33000, Bordeaux, France

**Abstract** Encoding continuous sensory variables requires sustained synaptic signalling. At several sensory synapses, rapid vesicle supply is achieved via highly mobile vesicles and specialized ribbon structures, but how this is achieved at central synapses without ribbons is unclear. Here we examine vesicle mobility at excitatory cerebellar mossy fibre synapses which sustain transmission over a broad frequency bandwidth. Fluorescent recovery after photobleaching in slices from VGLUT1$^{Venus}$ knock-in mice reveal 75% of VGLUT1-containing vesicles have a high mobility, comparable to that at ribbon synapses. Experimentally constrained models establish hydrodynamic interactions and vesicle collisions are major determinants of vesicle mobility in crowded presynaptic terminals. Moreover, models incorporating 3D reconstructions of vesicle clouds near active zones (AZs) predict the measured releasable pool size and replenishment rate from the reserve pool. They also show that while vesicle reloading at AZs is not diffusion-limited at the onset of release, diffusion limits vesicle reloading during sustained high-frequency signalling.

*For correspondence: nusser@ koki.hu (ZN); a.silver@ucl.ac.uk (RAS)

**Competing interests:** The authors declare that no competing interests exist.

## Introduction

At the early stages of auditory, vestibular and visual pathways, information is transmitted at high rates across specialized ribbon synapses (*de Ruyter van Steveninck and Laughlin, 1996*; *Matthews and Fuchs, 2010*). At these synapses, vesicular release is sustained by a large number of highly mobile vesicles (*Holt et al., 2004*; *Rea et al., 2004*) and a ribbon-like scaffold that is thought to rapidly capture and deliver vesicles to the release sites (*Griesinger et al., 2005*; *Khimich et al., 2005*; *LoGiudice et al., 2008*; *Graydon et al., 2014*). Downstream of ribbon synapses, sensory information is transmitted by conventional synapses formed by afferent and sensory nuclei neurons that sustain rate-coded signalling over a broad frequency bandwidth (*Taschenberger and von Gersdorff, 2000*; *Saviane and Silver, 2006*; *Bagnall et al., 2008*; *Hallermann et al., 2010*). Some of these conventional synapses, such as those formed by cerebellar mossy fibre terminals (MFTs) or vestibular nerve fibres, have a readily releasable pool (RRP) of only 1–2 vesicles per AZ, which are docked and primed and ready for release. A much larger releasable pool (RP) containing 200–300 vesicles resides nearby and these vesicles can be supplied to the AZ, docked and primed at a combined rate of 40–80 s$^{-1}$ (with all three steps referred together as reloading) to refill the RRP (*Saviane and Silver, 2006*; *Hallermann et al., 2010*; *McElvain et al., 2015*). Even when the large RP is depleted by sustained high-frequency stimulation, release rates of 7–8 s$^{-1}$ can be sustained by replenishment from a large vesicle reserve pool (R) at these synapses (*Saviane and Silver, 2006*;

*McElvain et al., 2015*). Because these conventional synapses lack ribbons, and central synapses are thought to contain mostly immobile vesicles (*Jordan et al., 2005*; *Lemke and Klingauf, 2005*; *Shtrahman et al., 2005*; *Lee et al., 2012*), it is unclear how vesicles are supplied to AZs during sustained high-frequency signalling.

Vesicles are actively transported within axons at ~1–5 μm/s (*Brown, 2003*), but passive diffusion is thought to dominate vesicle translocation within the presynaptic terminals of both ribbon (*Holt et al., 2004*; *Rea et al., 2004*; *LoGiudice et al., 2008*; *Graydon et al., 2014*) and conventional central synapses (*Tokuoka and Goda, 2006*). Presynaptic terminals are packed with macromolecules and cytoskeletal elements, as well as vesicles and mitochondria (*Harris and Weinberg, 2012*; *Wilhelm et al., 2014*). The rate of vesicle diffusion will therefore be determined not only by the cytoplasmic viscosity, but by the cytoskeletal matrix (*Luby-Phelps et al., 1987*) and organelles including vesicles (*Gaffield et al., 2006*), as well as binding to these elements (*Shtrahman et al., 2005*) via fine protein connectors (*Siksou et al., 2007*; *Fernández-Busnadiego et al., 2013*). These factors, together with the tortuous paths required to diffuse around large organelles, are thought to explain why vesicle mobility in crowded presynaptic terminals is substantially lower than that of vesicle-sized beads in cytoplasm (*Luby-Phelps et al., 1987*; *Gaffield et al., 2006*). Hydrodynamic interactions between vesicles are another potentially important determinant of their mobility. Hydrodynamic forces arise from the displacement of fluid as nanoscale objects move through solution. Studies of protein diffusion in red blood cells and bacteria (*Doster and Longeville, 2007*; *Ando and Skolnick, 2010*) show hydrodynamic interactions reduce the mobility of macromolecules. Moreover, studies of colloidal suspensions of vesicle-sized beads demonstrate hydrodynamic interactions are particularly strong in crowded environments, when the volume fraction of the diffusing objects is high (*van Blaaderen et al., 1992*; *Tokuyama and Oppenheim, 1994*; *Segrè et al., 1995*). However, nothing is known about how hydrodynamic interactions affect vesicle mobility within axon terminals. Quantifying the physical determinants of vesicle mobility within crowded presynaptic terminals could therefore provide new insights into vesicle supply at central synapses.

We have investigated the physical determinants of vesicle mobility in cerebellar MFTs, large central glutamatergic synapses that sustain broad-bandwidth rate-coded sensory signalling (*Saviane and Silver, 2006*; *Hallermann et al., 2010*). To do this, we combined fluorescent recovery after photobleaching (FRAP) in Venus-tagged VGLUT1 (VGLUT1$^{Venus}$) knock-in mice (*Herzog et al., 2011*) with serial-section electron microscopy (EM), electron tomography and 3D reaction-diffusion modelling of the presynaptic environment. Our results establish that most vesicles within MFTs are highly mobile and hydrodynamic interactions and vesicle collisions are major determinants of their mobility. Moreover, simulations of vesicle diffusion at 14 reconstructed AZs indicate that, while vesicle diffusion does not limit vesicle reloading at the onset of sustained high-frequency signalling, it does limit vesicle reloading at late times due to significant vesicle depletion near the AZ. The simulations also predict both the size of the functional RP and the vesicle replenishment rate from the reserve pool.

## Results

### FRAP of VGLUT1$^{Venus}$ labelled vesicles in cerebellar MFTs

We investigated vesicle mobility within cerebellar MFTs using FRAP (*Figure 1*) at near-physiological temperature (35°C) in acute slices obtained from young-adult (P22-33) VGLUT1$^{Venus}$ knock-in mice (*Herzog et al., 2011*). In the cerebellar input layer, VGLUT1 is expressed in MFTs arising from several pre-cerebellar nuclei (*Gebre et al., 2012*; *Hisano et al., 2002*). Whole-cell patch-clamp recordings from postsynaptic granule cells (GCs) confirmed EPSC amplitudes, kinetics and short-term plasticity of cerebellar MFT-GC synapses in VGLUT1$^{Venus}$ mice were similar compared to wild-type mice (*Figure 1—figure supplement 1*). Moreover, previous work showed the expression and subcellular distribution of VGLUT1$^{Venus}$ is indistinguishable from that of native VGLUT1 in wild-type mice (*Herzog et al., 2011*). The advantages of studying vesicle mobility using VGLUT1$^{Venus}$ knock-in mice rather than lipophilic dyes such as FM1-43 include the ability to use acute slices rather than primary cultures, and the capacity to monitor fluorescence of both mobile and immobile vesicles.

MFTs expressing VGLUT1$^{Venus}$ were visualized with conventional fluorescence microscopy (*Figure 1A*). To assay vesicle mobility, we positioned a diffraction-limited laser focal spot within a

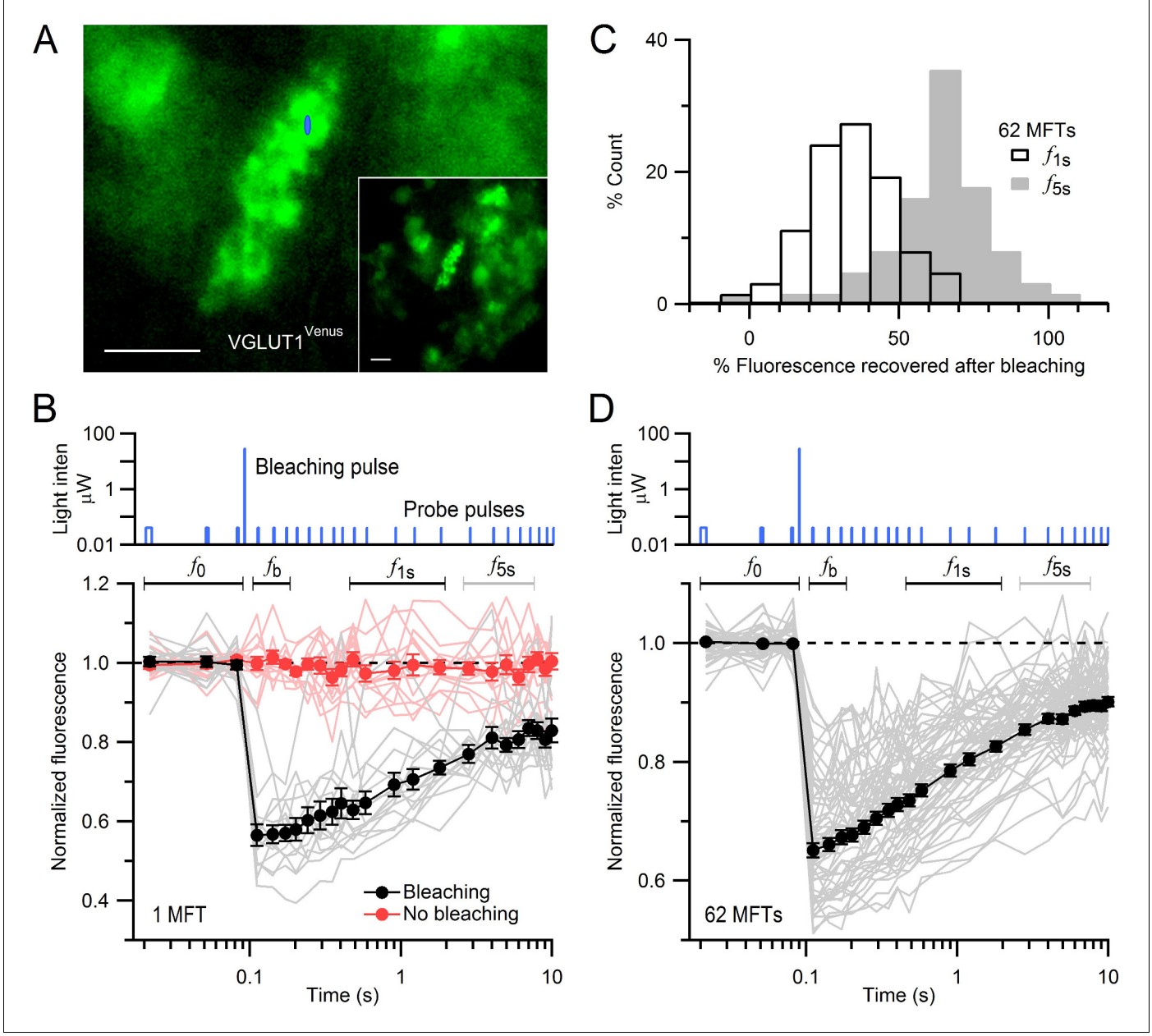

**Figure 1.** FRAP of vesicles in MFTs of VGLUT1[Venus] knock-in mice. (**A**) A VGLUT1[Venus]-labelled MFT near the surface of a cerebellar slice. Blue ellipse denotes *xz* dimensions of iPSF. Inset, lower magnification. Scale bars: 5 μm. (**B**) Fluorescence recovery after photobleaching (FRAP) measurements from 15 locations within a single MFT (bottom, gray lines; note logarithmic timescale) using 2-ms low-intensity laser probe pulses before and after a single 0.5-ms high-intensity laser bleaching pulse (top; note logarithmic *y*-scale). Fluorescence measured without the bleaching pulse in the same MFT from 14 random locations (red lines; interleaved with recordings with bleaching). Data were normalized to the average fluorescence of the first 3 probe pulses ($f_0$) before bleaching. Filled black and red circles are means ± SEM. Black and gray horizontal lines denote measurement windows for $f_{1s}$ and $f_{5s}$ reported in **C**. (**C**) Average fluorescence recovery for 62 MFTs at 1 s (black; $f_{1s}$, average 0.4–1.7 s) and 5 s (gray; $f_{5s}$, average 2.7–6.9 s) after bleaching, normalized between $f_0$ and $f_b$, where $f_b$ is the fluorescence just after bleaching (average 0.02–0.08 s). (**D**) Same as **B** but gray lines are averaged FRAP curves from 62 MFTs (3–24 recordings per MFT) and black circles are the weighted population average, computed by ***Equation (1)***.

The following figure supplement is available for figure 1:

**Figure supplement 1.** VGLUT1[Venus] mice show normal MFT-GC synaptic transmission.

MFT and recorded fluorescence using the confocal spot detection method. The intensity of a brief bleaching pulse was set to produce a modest reduction in fluorescence (~35%) to ensure the bleached volume did not extend significantly beyond the core of the illumination point-spread function of our microscope (iPSF; FWHM $xy$ = 0.30 μm, $z$ = 1.32 μm; $e^{-2}$ volume = 0.31 μm$^3$). Fluorescence was monitored before and after the bleaching pulse using brief low-intensity probe pulses that produced little cumulative bleaching (*Figure 1B*, red circles). Since the iPSF was considerably smaller than the MFTs (*Figure 1A*, blue spot), which are typically 7 × 10 μm, we made multiple FRAP recordings from several locations within the same MFT (*Figure 1B*). While the individual FRAP measurements were variable, fluorescence almost always exhibited a strong recovery within 10 s (grey lines) indicating unbleached and bleached vesicles were free to move in and out of the confocal volume. The mean fluorescence recovery was determined for each MFT by averaging the individual FRAP measurements (black circles). To determine whether fluorescence recovery varied between MFTs, we calculated the fluorescence at two times, 1 s ($f_{1s}$) and 5 s ($f_{5s}$), during the recovery. Distributions of $f_{1s}$ and $f_{5s}$ across 62 MFTs were unimodal with mean 35 ± 2 and 63 ± 2%, respectively (*Figure 1C*). We therefore calculated a population mean FRAP curve by performing a weighted mean across all 62 MFTs (*Figure 1D*). The time of half recovery ($t_{1/2}$) of the average FRAP curve was 0.8 s and 77% of the bleached fluorescence recovered within 10 s. These results indicate the majority of vesicles within MFTs are highly mobile.

## Modulation of vesicle mobility

To test whether our FRAP measurements reflected the movement of vesicles within MFTs, we performed several manipulations known to slow or speed vesicle mobility. Reducing the temperature from 35° to 21°C slowed the fluorescence recovery (*Figure 2A*; $f_{1s}$ = 35 ± 2 vs. 25 ± 3%, respectively, p<0.01; n = 62 vs. 36 MFTs). Moreover, the $Q_{10}$ of the $t_{1/2}$ was 1.5, which is closer to that expected for passive diffusion ($Q_{10}$ ~ 1.3) than for active transport in axons ($Q_{10}$ ~ 3; *Forman et al., 1977*). Disruption of the actin cytoskeleton with 10 μM cytochalasin-D and 10 μM latrunculin-B (n = 60 MFTs) sped the fluorescence recovery compared to control conditions ($f_{1s}$ = 45 ± 2 vs. 35 ± 2%, respectively, p<0.001). In contrast, 5 μM jasplakinolide (n = 44 MFTs), a peptide that stabilizes actin filaments in vitro (*Bubb et al., 2000*), slowed fluorescence recovery compared to control conditions ($f_{1s}$ = 27 ± 2 vs. 35 ± 2%, p<0.01). These results suggest actin filaments within MFTs (*Hirokawa and Yorifuji, 1989*) have a modest effect on vesicle mobility.

Application of okadaic acid (OA; n = 42 MFTs), a nonspecific phosphatase inhibitor, dramatically sped the fluorescence recovery (*Figure 2B*; $f_{1s}$ = 64 ± 5 vs. 35 ± 2%, p<0.001, unequal-variance *t*-test), consistent with other studies (see *Table 1*, legend; *Jordan et al., 2005*; *Shtrahman et al., 2005*; *Gaffield et al., 2006*). Moreover, OA decreased the immobile fraction from 25% to 10–15%, indicating the immobile fraction can be attributed to immobile vesicles rather than large immobile organelles.

To examine the potential role of synapsin in MFTs (*Hirokawa and Yorifuji, 1989*) we applied the Cdk5 inhibitor roscovitine (n = 45 MFTs), which has been shown to increase vesicle mobility in hippocampal synapses from wild-type but not synapsin-knockout mice (*Orenbuch et al., 2012*). We found 50 μM roscovitine had no detectable effect on the rate of fluorescence recovery (*Figure 2B*; $f_{1s}$ = 38 ± 3 vs. 35 ± 2%, p=0.3) suggesting that, as for ribbon synapses (*Mandell et al., 1990*), there is little synapsin-based vesicle clustering at the centre of MFTs.

Finally, we repeated our FRAP experiments using paraformaldehyde-fixed slices (n = 17 MFTs), where subcellular components, including vesicles, were immobilized by cross-linking, and found fluorescence recovery was nearly absent (*Figure 2B*). However, there was a small but consistent increase in fluorescence after 2 s due to slow tissue drift (*Figure 2—figure supplement 1A–C*). A simple linear correction for the drift allowed a more accurate estimate of the control FRAP curve (*Figure 2—figure supplement 1D*). Together, these results confirm our FRAP measurements reflect the diffusion of vesicles within MFTs, suggest vesicle mobility is reduced by the presence of a network of actin filaments in the cytoplasm, and indicate synapsin has little impact on vesicle mobility within the interior of MFTs.

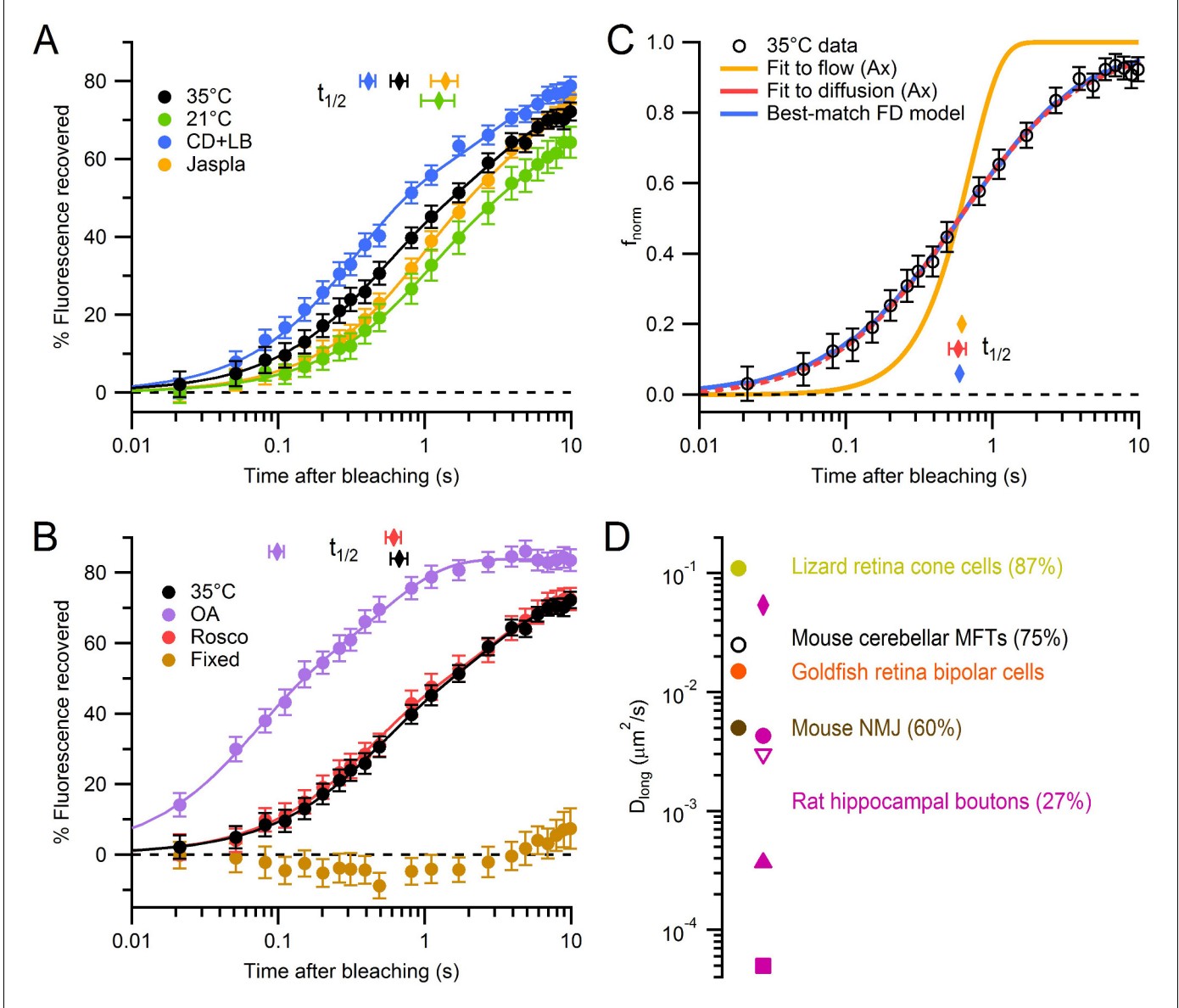

**Figure 2.** Modulation and quantification of vesicle mobility in MFTs. (**A**) Weighted average FRAP curve for control conditions (black circles; 35°C; 62 MFTs, 619 locations, 6 mice; from *Figure 1D*), 21°C (green circles; 36 MFTs, 414 locations, 2 mice), 10 μM cytochalasin-D and 10 μM latrunculin-B (blue circles; 60 MFTs, 458 locations, 3 mice) and 5 μM jasplakinolide (yellow circles; 44 MFTs, 492 locations, 2 mice). Lines show double-exponential fits. Diamonds indicate $t_{1/2}$. Data were normalized between $f_0$ and $f_b$, where $f_b$ was estimated from the fits at $t = 0$. (**B**) Same as **A** but for 2 μM okadaic acid (purple circles; 42 MFTs, 388 locations, 2 mice), 50 μM roscovitine (red circles; 45 MFTs, 374 locations, 3 mice) and fixed slices (brown circles; 17 MFTs, 168 locations, 1 mouse). (**C**) Fit of the analytical solution of Axelrod et al. (Ax; *Axelrod et al., 1976*) for passive diffusion ($D_{long}$ = 0.025 ± 0.003 μm²/s; red line) and directed flow ($V_o$ = 0.344 ± 0.012 μm/s; yellow line) to normalized drift-corrected control FRAP data (open black circles; *Figure 2—figure supplement 1D*). Blue line shows the best-match finite-difference (FD) simulation using iPSF and cPSF of our microscope, a 0.5 ms bleaching pulse, 2 ms probe pulses and $D_{long}$ = 0.028 μm²/s. (**D**) $D_{long}$ for MFTs (black), goldfish retina bipolar cells (orange; *Holt et al., 2004*), lizard retina cone cells (yellow; *Rea et al., 2004*) and mouse NMJ (brown; *Gaffield and Betz, 2007*). $D_{long}$ for rat hippocampal boutons at room temperature computed from fluorescence correlation spectroscopy (FCS) assuming a pure diffusion (pink circle), stick and diffuse model (diamond), a caged diffusion model (filled triangle; *Yeung et al., 2007*), a caged diffusion model using different FCS data (square, *Jordan et al., 2005*), and for single-vesicle tracking measurements (open triangle, *Lee et al., 2012*). Mobile fractions are given in parentheses if known (MFT value is from *Figure 4*)

The following source data and figure supplement are available for figure 2:

**Source data 1.** Average FRAP curves for single MFTs for various conditions.

**Source data 2.** Average percent fluorescence recovered at 1 s and 5 s after bleaching.

*Figure 2 continued on next page*

*Figure 2 continued*

**Source data 3.** Parameters file for best-match finite-difference FRAP simulation.

**Figure supplement 1.** Characterization of tissue drift and correction of FRAP curves.

## The long-time self-diffusion coefficient of vesicles in MFTs

Vesicle mobility is often quantified by calculating the 'effective' diffusion coefficient, also known as the long-time self-diffusion coefficient ($D_{long}$), which reflects the mobility of a diffusant on long time scales (*van Blaaderen et al., 1992*). We quantified $D_{long}$ of vesicles in MFTs by fitting our average control FRAP curve to an analytical solution of the diffusion equation (*Axelrod et al., 1976*). This gave $D_{long} = 0.018 ± 0.005$ μm$^2$/s for the raw control FRAP curve and $0.025 ± 0.003$ μm$^2$/s after correcting for tissue drift (*Figure 2C*, red line; *Figure 2—figure supplement 1D*; *Table 1*). In contrast to the diffusion model, which matched our data closely, the fit of a model of directed flow (*Axelrod et al., 1976*) was poor (*Figure 2C*, yellow line). Moreover, the fit of the weighted sum of the diffusion and flow models converged on diffusion (98%), suggesting diffusion underlies vesicle mobility in MFTs. Because some of the assumptions underlying the analytical diffusion model only approximated our experimental conditions, we also used 3D finite-difference reaction-diffusion simulations that explicitly modelled these conditions (Materials and methods). The FRAP simulation that best matched the drift-corrected data had a $D_{long}$ similar to that obtained with the analytical solution ($p=0.84$, F-test; *Figure 2C*, blue line). A comparison of our best estimate for $D_{long}$ in MFTs ($0.025$ μm$^2$/s) with that estimated at other synapses (*Figure 2D*) indicates vesicle mobility is higher in MFTs than at other conventional central synapses and the neuromuscular junction (NMJ), but comparable to that measured at ribbon synapses.

**Table 1.** Estimates of $D_{long}$ under various experimental conditions.

| Solution | °C | $t_{1/2}$ (s) | $D_{long}$ (μm$^2$/s) | % Recovered |
|---|---|---|---|---|
| STRD (-drift) | 35 | 0.58 ± 0.08 | 0.025 ± 0.003 | 67 |
| STRD | 35 | 0.81 ± 0.10 | 0.018 ± 0.005 | 77 |
| STRD | 21 | 1.39 ± 0.24 | 0.010 ± 0.002 | 73 |
| CD + LB (10 μM) | 35 | 0.46 ± 0.05 | 0.032 ± 0.003 | 80 |
| Jaspla (2 μM) | 35 | 0.96 ± 0.13 | 0.015 ± 0.002 | 76 |
| Jaspla (5 μM) | 35 | 1.41 ± 0.15 | 0.010 ± 0.001 | 84 |
| OA (2 μM) | 35 | 0.12 ± 0.02 | 0.120 ± 0.018 | 85 |
| Rosco (50 μM) | 35 | 0.72 ± 0.11 | 0.020 ± 0.003 | 77 |

STRD: Standard ACSF. -drift: data corrected for tissue drift (*Figure 2—figure supplement 1*); all other measurements are not drift corrected. CD + LB: 10 μM cytochalasin-D plus 10 μM latrunculin-B. Jaspla: jasplakinolide. OA: okadaic acid. Rosco: roscovitine. Values for $D_{long}$ and $t_{1/2}$ (± STDV) were computed by fitting experimental FRAP curves (*Figure 2A,B*) to *Equation (2)*.

The effect of OA on vesicle mobility in the MFT is in close agreement with that reported by *Shtrahman et al. (2005)* who report $D_{long} = 0.10$ μm$^2$/s for hippocampal boutons in OA. While our results do show a reduction in the immobile vesicle fraction, this reduction is not enough to account for the large increase in $D_{long}$. Instead, the increase in $D_{long}$ is more likely due to a reduction in protein interactions between the vesicles and cytoskeleton, as suggested by Shtrahman et al., in which case the effects of OA will be reflected in a change in $D_{cyto}$. Using data from *Figure 5*, we estimate $D_{cyto} = 0.515$ μm$^2$/s in OA, a four-fold increase from control conditions ($0.127$ μm$^2$/s).

**Source data 1.** Fits of Axelrod equation to FRAP curves.

## Quantification of organelle crowding within MFTs

Densely packed organelles within presynaptic terminals are expected to lead to crowding effects (*Wilhelm et al., 2014*) that could affect vesicle mobility (*Gaffield et al., 2006*). To quantify vesicle and mitochondrial densities within the central region of MFTs, where our FRAP measurements were predominantly made, we performed serial-section EM (*Figure 3A*). Quantitative analysis of 3D volumes (0.2–0.4 $\mu m^3$, $n$ = 3) revealed a vesicle density of 3930 ± 262 per $\mu m^3$ (or 118 ± 8 per $\mu m^2$ in 2D) in regions not occupied by mitochondria, consistent with previous measurements from MFTs (*Palay and Chan-Palay, 1974*). To convert this density into a volume fraction, we estimated the mean volume occupied by a vesicle by performing high-resolution 3D electron tomography on MFTs. Vesicle diameters exhibited an approximately normal distribution with a mean of 41.1 ± 0.2 nm (256 vesicles around 3 AZs; *Figure 3—figure supplement 1*), again consistent with previous measurements (*Palay and Chan-Palay, 1974*). Taking into account vesicle membrane proteins (*Takamori et al., 2006*), and the cubic relationship between vesicle radius and volume, indicates the mean volume occupied by a vesicle is equivalent to a sphere with 44 nm diameter. Using this diameter, we computed a vesicle volume fraction of 0.17 ± 0.01 of the non-mitochondrial volume (*Figure 3B*). The average volume fraction occupied by mitochondria was 0.28 ± 0.04. Hence, this analysis shows the central region of the MFT is a highly crowded environment with 40% of the volume occupied by vesicles and mitochondria.

## Effects of organelle crowding on vesicle mobility

To investigate the impact of organelle crowding on vesicle mobility in MFTs, we modelled vesicle diffusion in 3D using a hard-sphere Monte Carlo algorithm (*Cichocki and Hinsen, 1990*) that explicitly simulated the movement of individual vesicles and their collisions (*Figure 3C*). In these simulations, vesicle movement on each time step was determined by the short-time vesicle diffusion coefficient ($D_{short}$) which defines the rate of diffusion before collisions occur. Vesicles were not allowed to overlap with themselves or mitochondria. Since simulations mimicked conditions of live tissue, we corrected our measured vesicle diameter for the 11% tissue shrinkage in fixed tissue (*Korogod et al., 2015*) giving an equivalent diameter of 49 nm. The spatial extent of the bleaching reaction was set by the iPSF, and the average fluorescence during the probe pulses was computed by spatially weighting the fluorescence according to the measured confocal PSF (cPSF) of our microscope (*Figure 3C*, right; *Figure 3—figure supplement 2*). Approximately 300 vesicles were located within the cPSF (*Figure 3C*, left). When the timing and intensity of the bleaching reaction were set to match the bleaching and probe pulses used in our FRAP experiments, FRAP curves from the Monte Carlo simulations (*Figure 3D*) exhibited remarkably similar behaviour to our experimental FRAP curves (*Figure 1B*).

Only two parameters were varied in these experimentally constrained simulations: $D_{short}$ and the fraction of immobile vesicles. To find the values of these two parameters that produced a FRAP curve that best matched our experimental FRAP curve, we used a parameter search and chi-square ($\chi^2$) criterion (*Figure 4A,B*). The log of the $\chi^2$ calculated from the simulated and experimental FRAP curves had a minimum at $D_{short}$ = 0.060 $\mu m^2$/s and a 25% immobile fraction (68.3% confidence intervals 0.055–0.070 $\mu m^2$/s and 24–25%, respectively). To better understand how vesicle collisions reduce vesicle mobility, we examined how vesicle diffusion, computed from the mean square displacement (MSD) of the mobile vesicles, changed as a function of time (D($t$) = MSD/6$t$; *Figure 4C*). At short times, before any vesicle collisions, D($t$) ≈ $D_{short}$ (0.060 $\mu m^2$/s). At longer times, however, as vesicles collided with themselves and mitochondria, D($t$) fell to a steady-state value of 0.025 $\mu m^2$/s. Hence, over the 10 s period of our FRAP experiment, D($t$) reached a similar $D_{long}$ estimated from our analytical and finite-difference approaches (*Figure 2C*). Indeed, comparison of the FRAP curve computed from our best-match Monte Carlo simulation with that computed from our best-match finite-difference simulation showed a close match (*Figure 4D*). These results indicate steric interactions introduce a pronounced time dependence to vesicle diffusion and over long timescales vesicle collisions reduce vesicle mobility by a factor of 2.4 (i.e. $D_{long}/D_{short}$ = 0.42).

To understand how steric interactions affect vesicle mobility under different conditions, we explored the effect of different vesicle volume fractions, immobile vesicle fractions and mitochondria volume fractions on vesicle mobility. In the absence of immobile vesicles and mitochondria, increasing the vesicle volume fraction from 0 to 0.5 reduced vesicle mobility (measured as $D_{long}/D_{short}$) in a

near-linear manner (*Figure 5A*, red circles) as previously reported (black squares; *Cichocki and Hinsen, 1990*). Adding immobile vesicles further reduced the diffusion coefficient of the mobile vesicles (*Gaffield et al., 2006*) and as the fraction of immobile vesicles increased, this effect became increasingly nonlinear. Finally, adding mitochondria reduced vesicle mobility to a similar extent across all vesicle densities (blue open circles). For conditions within MFTs, the reduction in vesicle mobility due to collisions between mobile vesicles was 30%, while that due to collisions with immobile vesicles and mitochondria was 13% and 32%, respectively. These results show vesicle collisions are a major determinant of vesicle mobility at a central synaptic terminal.

## Effect of hydrodynamic interactions on vesicle mobility

Hydrodynamic interactions have long been known to be a major determinant of the self-diffusion of nanoscale beads in crowded colloidal suspensions (*van Blaaderen et al., 1992*; *Segrè et al., 1995*). Unlike steric interactions, hydrodynamic interactions occur on a very fast timescale, so their effect on vesicle diffusion can essentially be considered instantaneous. To estimate the effect of hydrodynamic interactions on vesicle diffusion in MFTs, we used an analytical approach that is accurate across a wide range of volume fractions (*Tokuyama and Oppenheim, 1994*). We quantified the effect of hydrodynamic interactions by calculating the ratio of $D_{short}$ to the 'free' unhindered diffusion coefficient of a single vesicle in cytoplasm ($D_{cyto}$). As shown in *Figure 5B*, $D_{short}/D_{cyto}$ shows an approximately linear relationship with vesicle volume fraction (red line). However, introduction of randomly dispersed immobile vesicles at 25% produced a nonlinear dependence between $D_{short}/D_{cyto}$ and the total vesicle volume fraction (blue line; *Freed and Muthukumar, 1978*; *Michailidou et al., 2009*). For the vesicle volume fraction found at the centre of MFTs, $D_{short}/D_{cyto} = 0.47$ (dashed blue line) suggesting hydrodynamic interactions reduce vesicle mobility by two-fold within MFTs.

## Combined effects of steric and hydrodynamic interactions on vesicle mobility

As hydrodynamic interactions act on a microsecond timescale, the vesicle diffusion coefficient $D(t)$ is expected to decrease from $D_{cyto}$ to $D_{short}$ almost instantaneously (*Figure 5C*). On the other hand, vesicle collisions take longer to occur, so their effect on $D(t)$ is expected to occur on a millisecond to second timescale. At these longer timescales, both hydrodynamic and steric interactions are therefore present. The different timescales of hydrodynamic and steric interactions suggest they can be treated as independent processes, allowing their combined effect to be calculated via a simple multiplication: $D_{long}/D_{cyto} = D_{short}/D_{cyto} \times D_{long}/D_{short}$ (*Figure 5D*; red circles and black line; *Medina-Noyola, 1988*; *van Blaaderen et al., 1992*). For the conditions at the centre of MFTs, the combined effect of hydrodynamic interactions and vesicle collisions resulted in an 80% reduction in vesicle mobility ($D_{long}/D_{cyto} = 0.19$; dashed blue line). This suggests the diffusion coefficient of dilute vesicles in cytoplasm, in the absence of crowding effects, is 0.127 $\mu m^2/s$, which is only a factor of 3.5 lower than that of dilute 50 nm beads in the cytoplasm of 3T3 cells at 37°C (0.45 $\mu m^2/s$; *Luby-Phelps et al., 1987*). A likely explanation of the lower value of $D_{cyto}$ for vesicles than beads is that vesicle protein interactions with the cytomatrix slow vesicle mobility.

## Quantification of the morphology around the AZ

To understand how hydrodynamic and steric interactions could influence vesicle supply at AZs, we quantified the morphological properties of 14 AZs using 3D reconstructions from high-resolution serial-section EM (*Figure 6A,B*). The AZ area, closely opposed to the postsynaptic density, varied across synaptic contacts (0.009–0.039 $\mu m^2$) with a mean of 0.017 ± 0.002 $\mu m^2$ (n = 14). This is smaller than most other central synapses which can have 10-fold larger areas (*Xu-Friedman and Regehr, 2004*; *Harris and Weinberg, 2012*; *Holderith et al., 2012*). When expressed as the diameter of a circular disk, our AZ measurements correspond to a range of 108–222 nm, with mean 145 ± 8 nm.

To determine the spatial extent of vesicles near the AZ, we designated those regions in the 3D reconstructions that contained vesicles as freely diffusible space, and all other regions as non-diffusible space (Materials and methods). We refer to the region of diffusible space extending from each AZ face as the vesicle 'cloud'. Vesicle density within the vesicle cloud was calculated either by counting the number of vesicle centre points falling within its boundaries, or by computing the volume fraction occupied by the vesicles (*Figure 6C*, bottom; red circles and blue line respectively). The

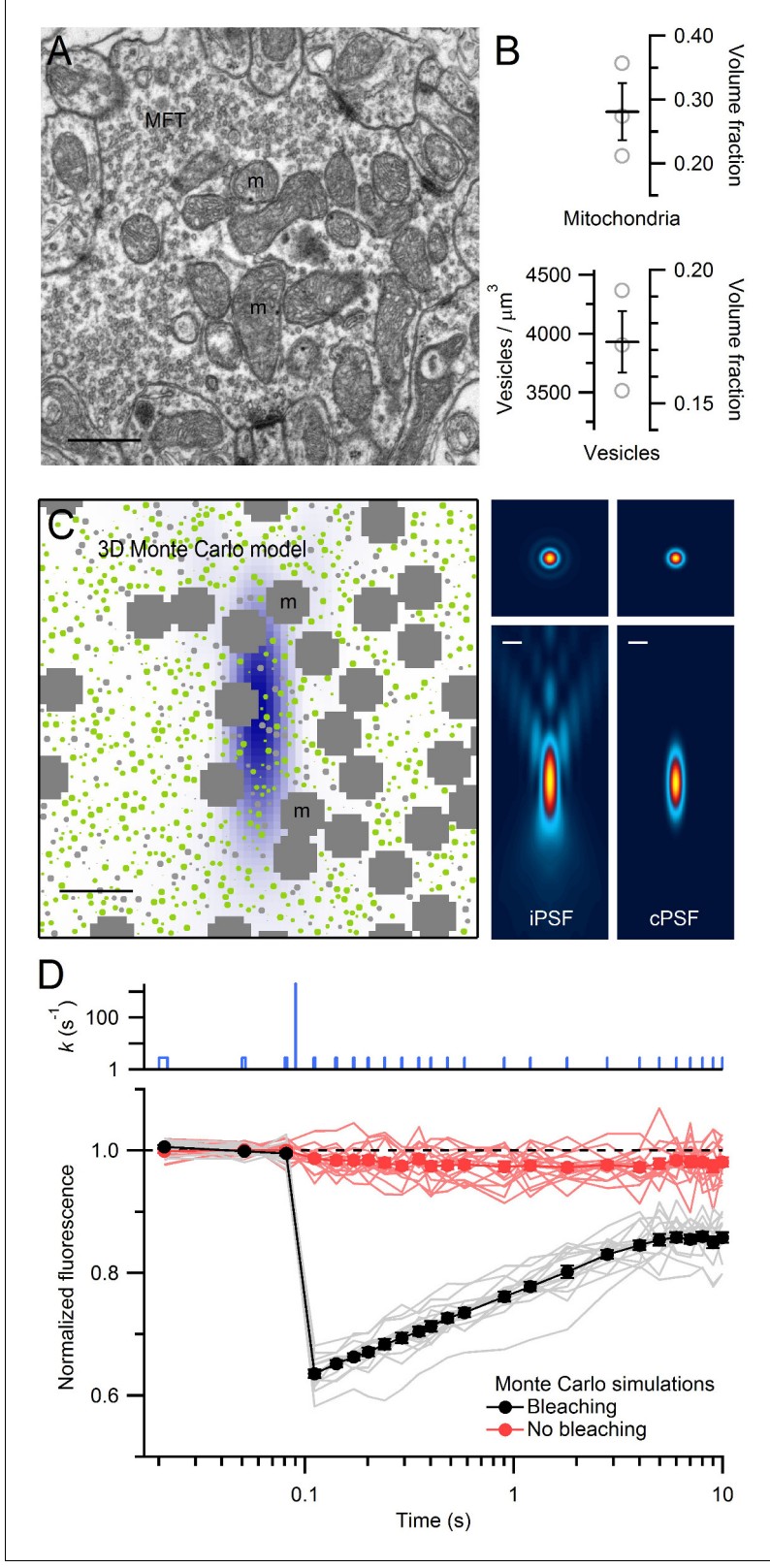

**Figure 3.** EM measurements of vesicle and mitochondrial densities within MFTs and Monte Carlo simulations of FRAP experiments. (**A**) Electron micrograph of a cerebellar MFT from adult mouse showing vesicles and mitochondria (m). Scale bar: 0.5 μm. (**B**) Mean density of vesicles and mitochondria (black lines) computed from electron micrographs from 3 MFTs (gray circles), where the vesicle density is computed for the non-mitochondrial

*Figure 3 continued on next page*

*Figure 3 continued*

volume. Vesicle volume fraction was computed assuming a diameter of 44 nm in fixed tissue (*Figure 3—figure supplement 1E*). (**C**) Left: *xz* cross section (3 × 3 µm) through the 3D Monte Carlo model of the MFT simulating live tissue conditions, showing randomly placed 49 nm vesicles (0.17 volume fraction) that are mobile (green) or immobile (light gray, 25%), and clusters of mitochondria (dark gray, 0.28 volume fraction). Differences in vesicle diameters reflect their different cross sections in a single plane. Blue shading denotes iPSF. Right: *xy* (top, 3 × 3 µm) and *xz* (bottom: 3 × 7 µm) cross sections of iPSF and cPSF (*Figure 3—figure supplement 2*). Scale bars: 0.5 µm. (**D**) FRAP simulations for model in C with (black) and without (red) the bleaching pulse, showing individual trials (lines) and averages (filled circles). Top: bleaching rate (*k*) of *Equation (4)* used for probe and bleaching pulses.

The following source data and figure supplements are available for figure 3:

**Source data 1.** Density of vesicles and mitochondria.
**Figure supplement 1.** Ultrastructure of VGLUT1-Venus expressing MFTs and measurements of vesicle diameter.
**Figure supplement 1—source data 1.** Synaptic vesicle diameters.
**Figure supplement 2.** Quantification of emission and confocal point spread functions.

vesicle density reached 5652 ± 377 per $µm^3$ at 30 nm from the AZ. This corresponds to a volume fraction of 0.25, assuming a vesicle diameter of 44 nm in fixed tissue, which is 50% higher than the volume fraction at the center of MFTs (0.17). Vesicle density declined with distance from the AZ, converging to the density estimated at the centre of the MFT for distances >100 nm. These results suggest an accumulation of vesicles extends ~2 vesicle diameters from the AZ. The total number of vesicles increased monotonically with distance from the AZ, from 2 ± 1 within 22 nm, to 48 ± 6 within 100 nm and 252 ± 34 within 300 nm (*Figure 6C*, top). These results show that, while there are few docked vesicles at the AZ, hundreds of vesicles reside nearby.

The similarity in vesicle density within the cloud and in the MFT interior (*Figure 6C*, bottom; red circles and dashed horizontal line, respectively), plus the irregular shape of the vesicle clouds, suggests diffusional barriers imposed by intracellular organelles and/or the plasma membrane are present in the AZ region. To examine this, we quantified the spatial dependence of the vesicle density assuming all of the space surrounding the AZ is diffusible (*Figure 6C*, bottom; black circles). This approach gave a similar estimate of the vesicle density close to the AZ, but beyond 150 nm from the AZ the density rapidly declined, reaching 15% of the peak value by 400 nm from the AZ. This suggests that, beyond 150 nm, intracellular organelles, cytoskeletal barriers and/or curvature of the plasma membrane create a significant amount of non-diffusible space (*Figure 3A*). Indeed, quantification of the mean distance from the AZ to the nearest mitochondrial membrane revealed mitochondria are located within 270 ± 20 nm (n = 29) of the AZ. These results suggest that the diffusible space available to vesicles is restricted near the AZ of MFTs and that vesicle density only increases above that found in the MFT interior within 100 nm of the AZ release face.

## Models of diffusion-limited vesicle supply to the AZ during sustained release

To examine the effectiveness of diffusion in supplying vesicles to AZs, we modelled vesicle diffusion in the vicinity of the 14 reconstructed AZs using our measured 3D geometries (*Figure 7A*). We assumed $D_{cyto}$ was identical to that measured in the centre of the MFT and regions outside the vesicle cloud consisted of non-diffusible space since these regions contained no vesicles in the EM reconstructions. The most distal regions of the vesicle cloud were coupled to a large reserve of vesicles to mimic replenishment from the centre of the MFT. Hydrodynamic interactions were simulated using analytical expressions that accounted for local variations in vesicle density and the effect of the plasma membrane, which reduced vesicle mobility as vesicles approached the AZ (*Figure 7—figure supplement 1*). In our initial simulations, vesicles that collided with the AZ were instantaneously 'released' without any delay. Under these conditions the vesicle release rate equals the

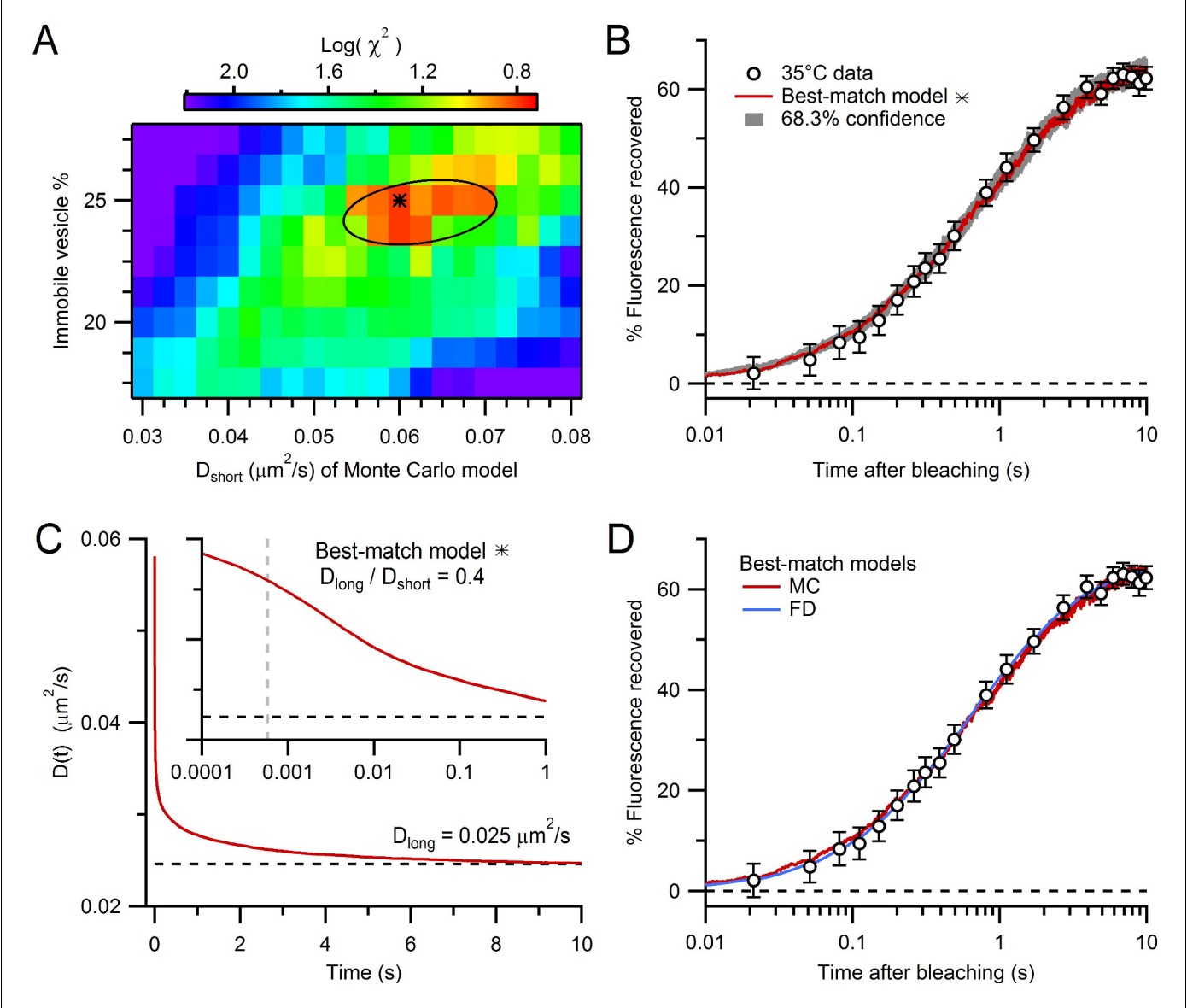

**Figure 4.** Estimation of vesicle diffusion coefficients $D_{short}$ and $D_{long}$. (**A**) Parameter search for the best match between the average drift-corrected control FRAP data (*Figure 2C*) and Monte Carlo (MC) simulations (*Figure 3C,D*) across a range of $D_{short}$ and % immobile vesicles, expressed as $\log(\chi^2)$. Black star denotes smallest $\chi^2$ ($D_{short}$ = 0.060 µm²/s, 25% immobile vesicles). Ellipse denotes 68.3% confidence region for two degrees of freedom ($\chi^2$ < 2.30). The vesicle step size ($dr$ = 2 nm) was sufficiently small to avoid discretization error and the simulation space (a 2 µm cube) was sufficiently large to avoid boundary effects (*Figure 4—figure supplement 1*). (**B**) Best-match simulation (red) compared to control FRAP data (open circles). Gray denotes 68.3% confidence. (**C**) D($t$) for best-match conditions in **A** with steady-state value ($D_{long}$ = 0.025 µm²/s; black dashed line) computed from a double-exponential fit for $t$ > 10 ms. Inset, D($t$) on a logarithmic timescale with average time to first collision (gray dashed line, 0.46 ms) when steric interactions start to reduce vesicle mobility. (**D**) Same as **B** but with added best-match finite-difference (FD) simulation with $D_{long}$ = 0.028 µm²/s (blue). $\log(\chi^2)$ = 0.8 (MC) and 0.5 (FD).

The following source data and figure supplement are available for figure 4:

**Source data 1.** Parameters file for best-match Monte-Carlo FRAP simulation.

**Figure supplement 1.** Comparison of Monte Carlo FRAP curves for different vesicle step size and simulation cube size.

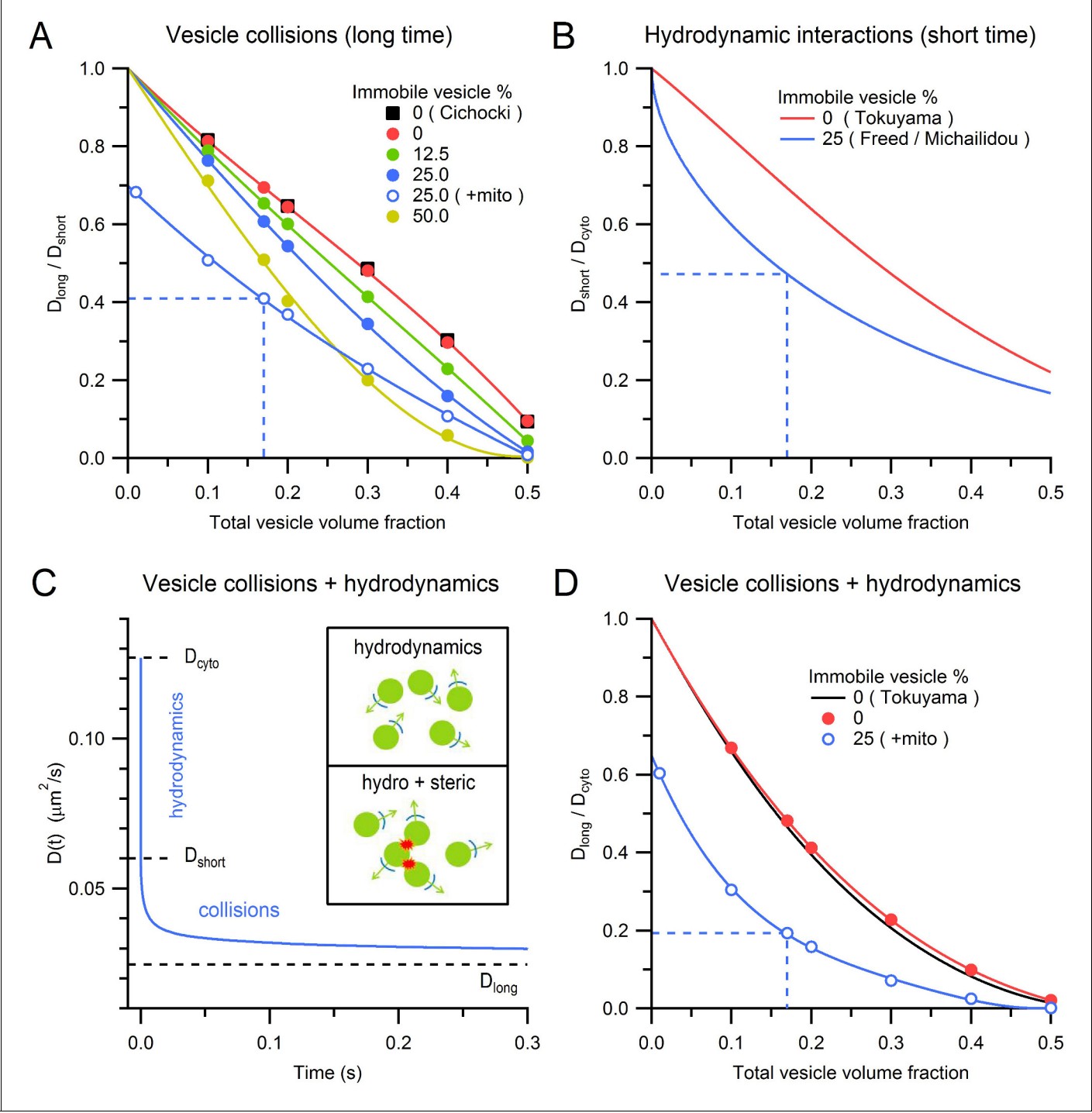

**Figure 5.** Effects of steric and hydrodynamic interactions on vesicle mobility. (**A**) Effect of steric interactions on $D_{long}$, normalized to $D_{short}$, as a function of the vesicle volume fraction and % immobile vesicles. In the absence of immobile vesicles, the results matched those of Cichocki and Hinsen (black squares; 1990). Also shown is $D_{long}/D_{short}$ for a 0.28 mitochondria volume fraction and 25% immobile vesicle fraction (blue open circles), with dashed blue line denoting average conditions at the centre of MFTs (0.17 vesicle volume fraction in the non-mitochondrial volume). $D_{long}$ was computed for an infinitely small vesicle step size ($dr = 0$) via linear extrapolation (**Figure 5—figure supplement 1**). Lines are polynomial fits. Error bars are smaller than symbols. (**B**) The effect of hydrodynamic interactions on $D_{short}$, normalized to $D_{cyto}$, as a function of the vesicle volume fraction for conditions when all vesicles are mobile (red line; **Equation 5**; **Tokuyama and Oppenheim, 1994**) or when 25% are immobile (blue line; **Equations 6,7**). (**C**) $D(t)$ for MFT conditions in **A** showing initial value ($D_{cyto}$), the reduction due to hydrodynamic interactions ($D_{short}$) and to both hydrodynamic and steric interactions ($D_{long}$). Inset, schematic diagram of hydrodynamic interactions between vesicles (top) and a combination of hydrodynamic and steric interactions (bottom). (**D**) Combined effect of steric and hydrodynamic interactions on $D_{long}/D_{cyto}$ as a function of vesicle volume fraction when all vesicles are

*Figure 5 continued on next page*

*Figure 5 continued*

mobile (red circles; computed via multiplication of data in **A** with data in **B**) compared to the theoretical prediction of Tokuyama and Oppenheim (black line; *Equation 8*). Blue circles denote the same MFT conditions as in **A**. Red and blue lines are polynomial fits.

The following figure supplement is available for figure 5:

**Figure supplement 1.** Monte Carlo estimate of D($t$) for an infinitely small vesicle step size.

rate at which vesicles are supplied to the AZ via diffusion, which we here simply call the vesicle supply rate.

At the earliest simulation times (0–2 ms) the vesicle supply rate varied between 25–4000 s$^{-1}$, with a mean of ~1000 s$^{-1}$ (*Figure 7B–D*). The average vesicle supply rate declined to ~200 s$^{-1}$ by 10 ms when ~5 vesicles had been released (*Figure 7F*). These values are well above the experimental vesicle reloading rates of 40–80 s$^{-1}$ per functional release site estimated at MFT-GC synapses during brief bursts of high-frequency stimulation (*Saviane and Silver, 2006*; *Hallermann et al., 2010*) and obtained during continuous depolarization of the presynaptic terminal (*Ritzau-Jost et al., 2014*). Even when vesicles were accumulated close to the AZ by simulating the effects of vesicle-vesicle binding due to fine protein connectors (*Figure 7—figure supplement 2*; *Siksou et al., 2007*; *Fernández-Busnadiego et al., 2013*), which reduced vesicle mobility by five-fold close to the AZ (*Figure 7—figure supplement 1*), the maximal vesicle supply rates remained higher than the experimental vesicle reloading rates (*Figure 7D*). These results suggest that at early times vesicle diffusion does not limit vesicle reloading at the AZ.

At intermediate times (~100 ms), when ~15 vesicles had been released, vesicle supply rates to the AZ approached the experimental vesicle reloading rates (*Figure 7C,E,F*), raising the possibility vesicle reloading could be limited by diffusion during sustained release. To investigate this, we focussed on later times (50–100 s) when the cumulative number of vesicles supplied to the AZ increased linearly with time, indicating the supply rate was constant (*Figure 7G*, blue line). The slope of the linear portion of the cumulative supply functions gave a mean vesicle supply rate of 7 ± 1 s$^{-1}$ (*Figure 7G, H*) which is remarkably similar to the vesicle release rates per release site measured at MFTs (8 s$^{-1}$; *Saviane and Silver, 2006*) and vestibular nerve synapses (7 s$^{-1}$; *McElvain et al., 2015*) during long trains of stimuli. Moreover, the back-extrapolation method for estimating the size of the pool of vesicles that can be depleted during the train (i.e. RRP+RP) gave an average of 237 ± 40 vesicles (*Figure 7I*), similar to values obtained for EPSC trains at MFTs (300; *Saviane and Silver, 2006*) and vestibular nerve synapses (200; *McElvain et al., 2015*). The close agreement between our experimentally constrained models of vesicle diffusion at AZs and experimental measurements suggests vesicular release is diffusion-limited during sustained high-frequency signalling once the RRP and RP are depleted. Examination of vesicle density close to the AZ during sustained release revealed that it falls dramatically, reaching a steady state after ~20 s (*Figure 7—figure supplement 1*, control conditions). However, vesicle mobility increased substantially during vesicle depletion due to reduced hydrodynamic and steric interactions, resulting in a shallow concentration gradient and a constant vesicle supply rate at late times.

To better understand the role of physical factors in vesicle supply to MFT AZs, we first removed hydrodynamic interactions from our simulations. This increased the vesicle supply rate at early, intermediate and late times (*Figure 7C–E,H*), indicating fast hydrodynamic interactions between vesicles play a key role in slowing vesicle supply to the AZ. Interestingly, removing steric interactions (by shrinking vesicles to points) had little additional effect to removing hydrodynamic interactions at early and intermediate times (*Figure 7C–E*). However, at late times, removing vesicle collisions increased vesicle supply to the AZ by a factor of 2 (*Figure 7C,G,H*) consistent with their effect on vesicle mobility over long time scales (*Figure 5D*). These results show the physical interactions arising from vesicle crowding limit the maximal vesicle supply rate to the AZ.

Next, we examined how the restricted diffusible space in the vicinity of the AZ affected vesicle supply. To do this we 'opened up' the 14 AZ geometries by removing the non-diffusible space surrounding each AZ and filled it with vesicles at the density measured at the centre of MFTs, thereby combining the RP and reserve pools (*Figure 7—figure supplement 2*). For open geometries, the average vesicle supply rate at early times was the same as that for the control geometries

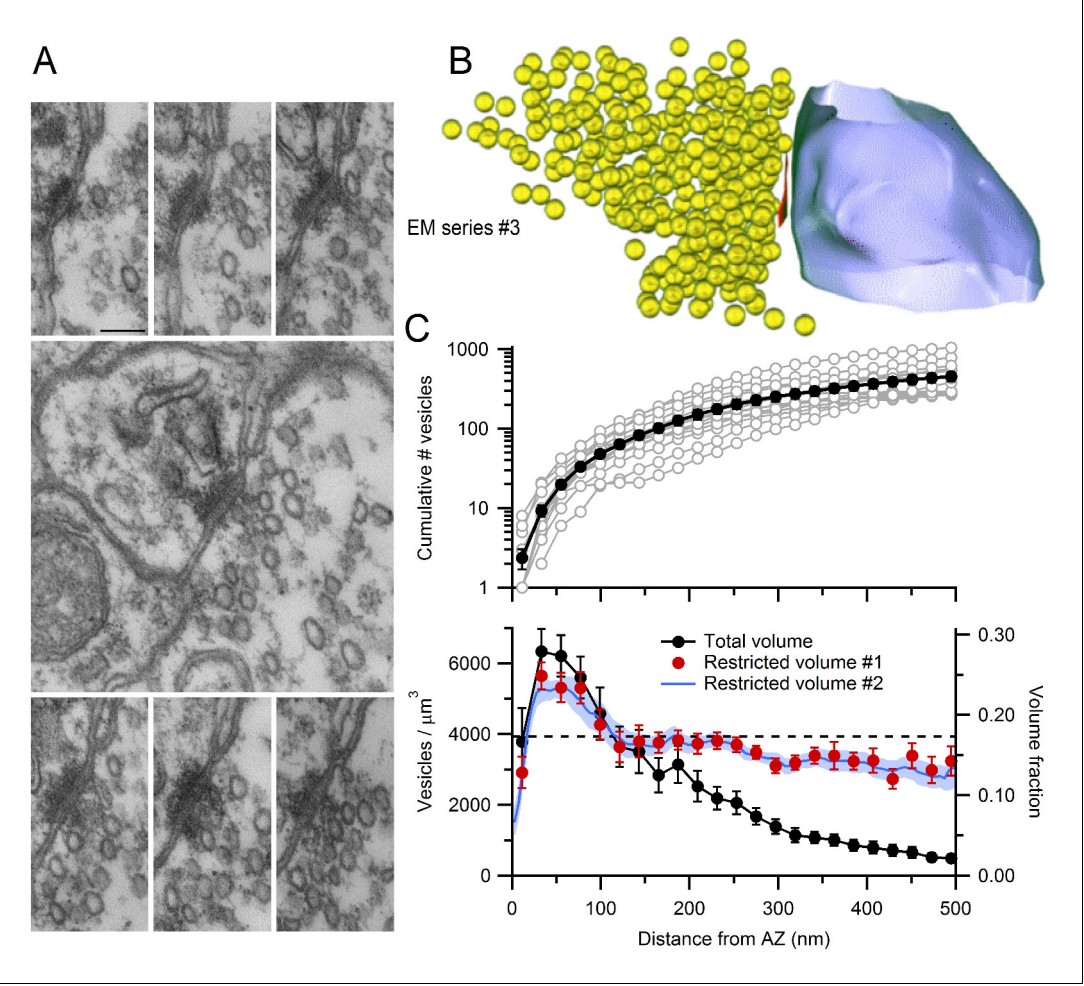

**Figure 6.** EM measurements of vesicle density near MFT AZs. (**A**) Serial-section electron micrographs containing a cerebellar MFT-GC synaptic junction (EM series #3). Scale bar: 100 nm. (**B**) 3D reconstruction of the synapse in **A** showing an AZ (red), synaptic vesicles (yellow) and postsynaptic GC dendrite (blue). (**C**) Vesicle count (top) as a function of distance from the AZ computed from 3D reconstructions as in **B** (n = 14; 22 nm bins). Counts of 0 for first bin are not shown (n = 3). Vesicle density (bottom; left axis; count per volume) for the total volume surrounding the AZ (black) or restricted volume within the vesicle cloud (red circles), and the vesicle volume fraction within the vesicle cloud computed using 4.4 nm voxels (blue line; right axis). Dashed line denotes vesicle density measured at the centre of the MFT (*Figure 3B*).

The following source data is available for figure 6:

**Source data 1.** AZ area and vesicle densities from 3D AZ reconstructions.

(*Figure 7C*). However, after ~1 s, the average vesicle supply rate for the open geometries levelled off to 55 s$^{-1}$, resulting in a seven-fold higher supply rate than the control (*Figure 7G,H*). Back extrapolation produced a pool size estimate of only 16 ± 3 vesicles (*Figure 7I*). These results suggest the presence of diffusion boundaries arising from membrane invaginations, intracellular organelles and possibly the actin cytomatrix (*Sankaranarayanan et al., 2003*; *Guillet et al., 2016*) limit diffusion-mediated vesicle supply to the AZ. Moreover, by determining the limiting vesicle supply rate during sustained release and thus the size of the pool of vesicles that can be depleted before release and supply rates equalize (excluding those replenished during that time), the geometry of the diffusible space close to the AZ determines the size of the functionally defined RP.

To examine how vesicle docking and priming (molecular and positional) and release probability affect our conclusions, we simulated 100 Hz MFT spike trains (*Figure 8*) assuming 1 or 2 release sites

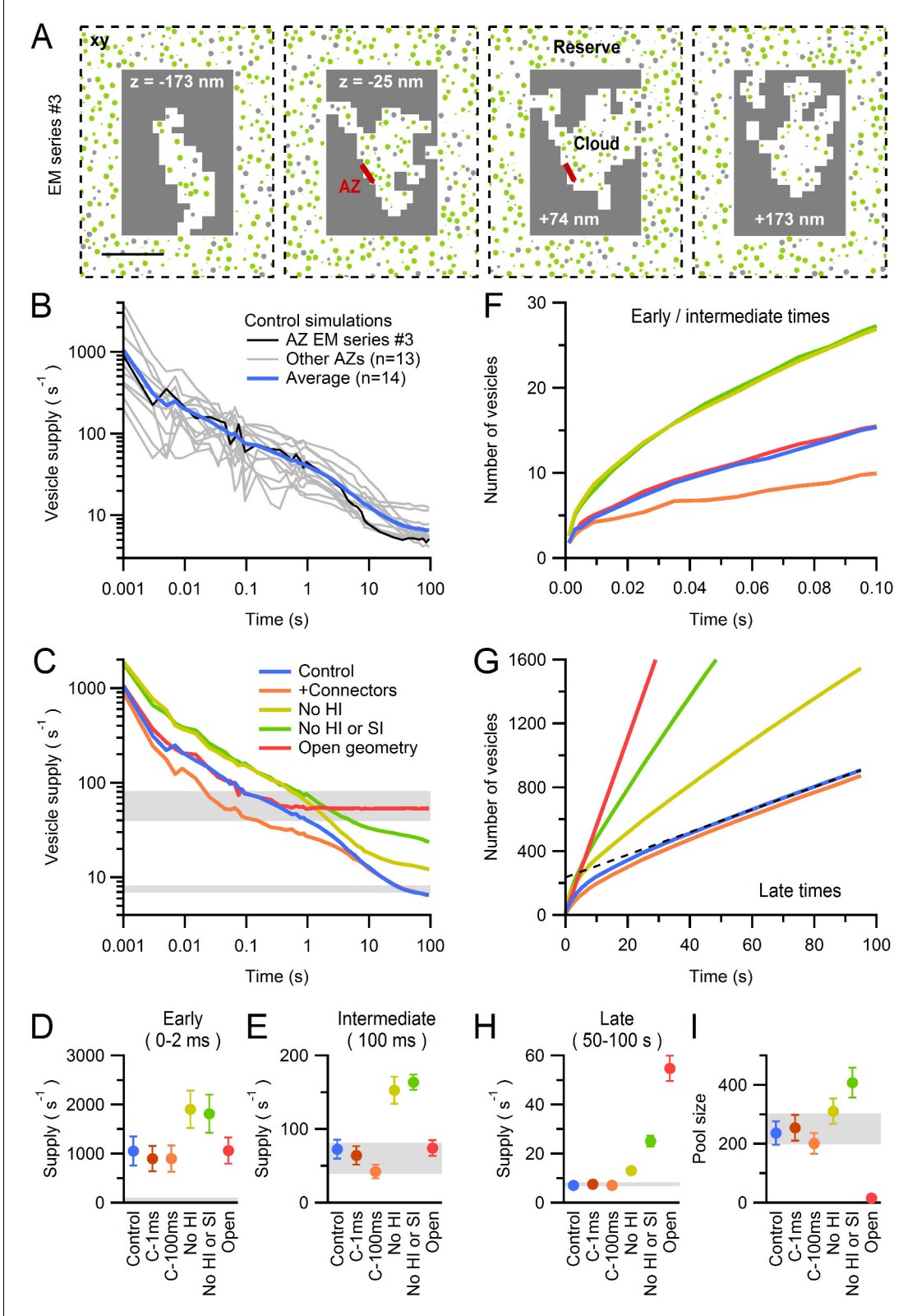

**Figure 7.** Diffusion-mediated vesicle supply to 14 MFT AZs. (**A**) *xy* cross sections through a Monte Carlo simulation of a 3D AZ reconstruction (*Figure 6B*; EM series #3) showing non-diffusible space (gray) surrounding the vesicle cloud and AZ (red), and reserve vesicles surrounding the cloud with restricted access. The immobile vesicle fraction was 25% (gray circles). For mobile vesicles within the cloud (green circles) $D_{short}$ was computed via a local density measurement, and hydrodynamic interactions arising from the plasma membrane (*Equations 9 and 10*) reduced $D_{short}$ as vesicles

*Figure 7 continued on next page*

*Figure 7 continued*

approached the AZ (*Figure 7—figure supplement 1B*). Vesicles that touched the AZ were instantaneously released. Only the central part of the simulation is shown. Scale bar: 0.5 μm. (B) Vesicle supply rate to the AZ for 14 AZ reconstructions (*Figure 6*) and their average (blue line). Black line denotes AZ in A. (C) Average control in B compared to the same simulations repeated for: vesicle-to-vesicle connectors and vesicle-to-AZ tethers with 100-ms lifetime for vesicles <150 nm of the AZ (orange; C-100 ms), no hydrodynamic interactions (No HI; yellow), no hydrodynamic or steric interactions between vesicles (No HI or SI; green; vesicles were simulated as dimensionless points; $D_{long} = D_{cyto} = 0.127$ μm$^2$/s) and an 'open' geometry where the vesicle cloud is continuous with the reserve (red). See *Figure 7—figure supplement 2*. (D,E) Average supply rate between 0–2 ms and at 100 ms computed for conditions in C and for connectors and tethers with 1 ms lifetime (C-1 ms). (F) Cumulative number of vesicles supplied to the AZ for simulations in C. (G) Same as F but for 100 s. Line fit to the control (black dashed line; 50–100 s) illustrates back extrapolation used to compute the RP size and the vesicle supply rate from the slope. (H,I) Supply rate between 50–100 s and pool size (as illustrated in G) for conditions in C. Estimates for an infinitely small vesicle step ($dr = 0$) for control conditions are similar to those shown here for $dr = 5$ nm (*Figure 7—figure supplement 3*). Gray shaded regions denote range of experimentally measured values.

The following source data and figure supplements are available for figure 7:

**Source data 1.** Vesicle supply rates and pool sizes computed from Monte Carlo AZ simulations.
**Source data 2.** Parameters file for one Monte-Carlo AZ simulation of EM series #3.
**Figure supplement 1.** Predicted vesicle mobility near the AZ.
**Figure supplement 2.** Three Monte Carlo simulation configurations for the 3D AZ reconstruction of EM series #3.
**Figure supplement 3.** Estimate of Monte Carlo AZ simulations for an infinitely small vesicle step size.

per AZ and a release probability of 0.5 (*Saviane and Silver, 2006*; *Hallermann et al., 2010*). To examine the limiting case, we assumed the total time for docking and priming ($\tau_{d+p}$) was equal to the measured RP→RRP vesicle reloading time at these AZs (16.7 ms; *Figure 8—figure supplement 1*). As expected, the initial release rate was lower than that for conditions of instantaneous release and, after the RRP was released, the vesicle release rate became limited by $\tau_{d+p}$ and the release probability (*Figure 8A–C*). However, at later times during the train (>1 s), the release rate converged with that for instantaneous release, indicating that it had become limited by the diffusion-mediated vesicle supply (*Figure 8A*). Back extrapolation analysis of cumulative release at late times during the 100 Hz train gave estimates for the limiting vesicle release rate and RP size that were similar to that obtained with instantaneous release, irrespective of whether 1 or 2 release sites were present (*Figure 8D,F,G*). These more realistic simulations of release during 100 Hz trains therefore support our finding that vesicle release becomes diffusion-limited during sustained high-frequency signalling. Furthermore, back extrapolation analysis of the open-geometry condition revealed that the release rate rapidly becomes limited to ~30 s$^{-1}$, resulting in a RP pool of only 2 ± 1 vesicles (i.e. close to the RRP; *Figure 8E–G*). While the exact values of the functional pool estimated from back extrapolation of the open-geometry condition may be prone to some error, because of its small size and the fact that the replenishment rate is higher than the release rate (*Neher 2015*), the more than 100-fold reduction in the RP estimate compared to that for measured vesicle clouds shows that the diffusible space around the AZ is a major determinant of the RP pool size.

## Discussion

We have investigated the physical determinants of vesicle mobility and supply within a large central mammalian axon terminal. Our results show 75% of vesicles within cerebellar MFTs are highly mobile, and hydrodynamic interactions and vesicle collisions arising from organelle crowding are major determinants of vesicle mobility. 3D reconstructions of 14 AZs from high-resolution serial-section EM revealed 'clouds' of vesicles near the AZ exhibiting variable shapes and sizes. Simulations of vesicle diffusion at these reconstructed AZs suggest that, at early times during high-frequency pre-synaptic activity, diffusion-mediated vesicle supply is substantially faster than the experimentally measured vesicle reloading rates. However, at later times, during prolonged presynaptic activity, when the RP becomes depleted, the rate of vesicular release becomes limited by diffusion-mediated

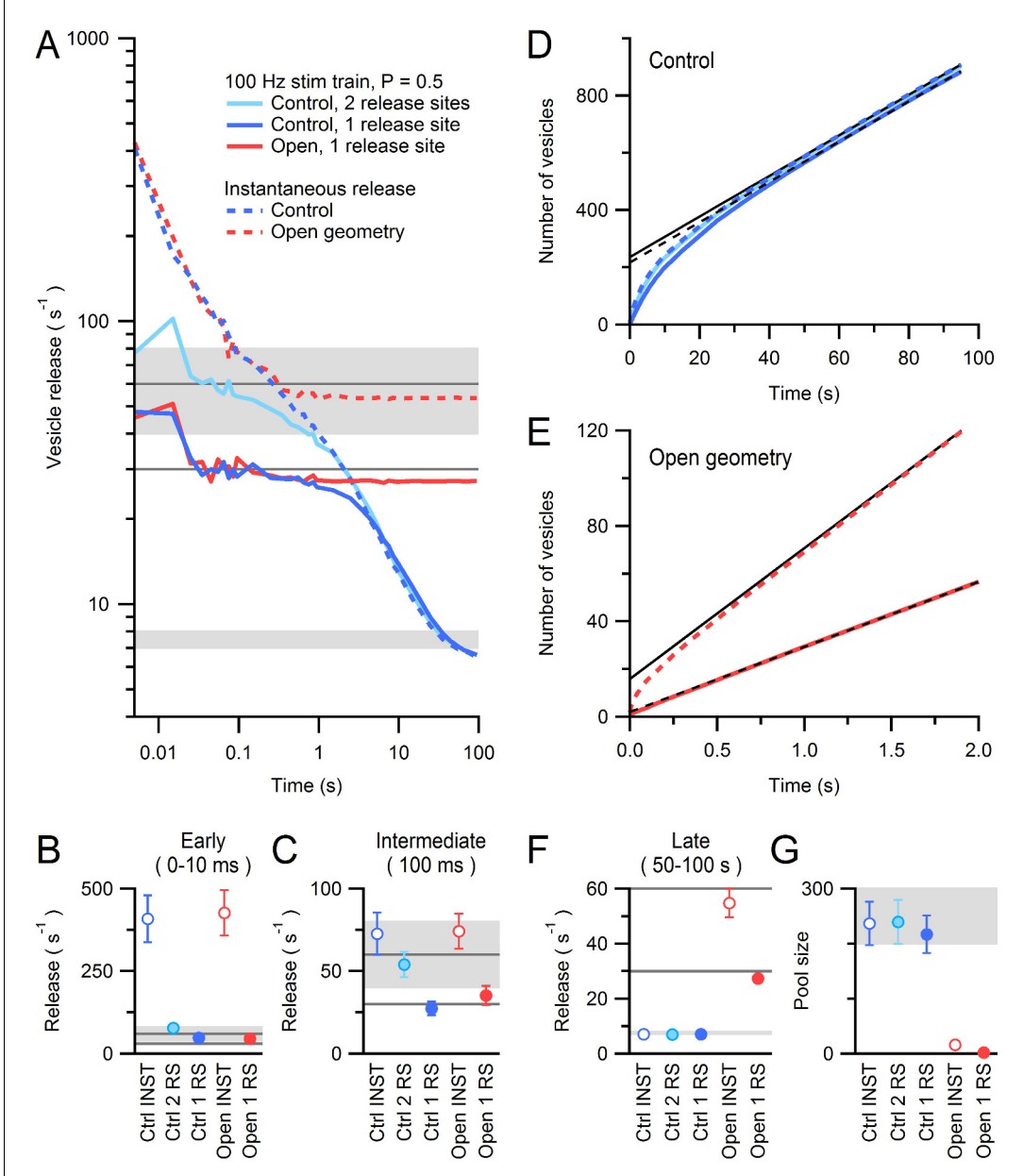

**Figure 8.** AZ simulations of release during a 100 Hz stimulus train with vesicle docking, priming and stochastic release. (A) Average vesicle release rate during simulations of a 100 Hz stimulus train across 14 AZs (control geometries) each with an RRP of 1 (dark blue line) or 2 (light blue line) vesicles (i.e. 1 or 2 release sites) and a release probability of 0.5. Reloading of the RRP was mediated by diffusion followed by a delay of 16.7 ms to account for docking/priming ($\tau_{d+p}$; *Figure 8—figure supplement 1*). Diffusion-limited vesicle supply rate to the AZ is shown for comparison (dashed blue line; instantaneous release condition from *Figure 7C*). Dark gray lines denote the maximal steady-state release rates (after the initial release of the RRP) for 1 or 2 release sites (30 and 60 s$^{-1}$, respectively); rates falling below these lines indicate vesicle diffusion is limiting vesicle release. Red line shows release from an AZ with an RRP of 1 and the same $\tau_{d+p}$ for open geometry conditions during a 100 Hz train. Diffusion-limited vesicle supply rate to the open geometry AZ is shown for comparison (dashed red line; instantaneous release condition from *Figure 7C*). (B) Average release rate between 0–10 ms computed for conditions in A (note difference in window length of early release compared to *Figure 7D* due to 10 ms inter-stimulus intervals). Open symbols denote instantaneous release conditions (INST). RS: release site. (C) Average release rate at 100 ms computed for conditions in A. At these times the release rates are limited by $\tau_{d+p}$ and the release probability (horizontal dark gray lines for 1 and 2 release sites), not by vesicle diffusion. (D) Cumulative number of vesicles released during 100 Hz train for control conditions (solid blue lines). At late times the simulations of stochastic release during the train overlap with those for instantaneous release (dashed blue line) and have similar slopes (dashed and solid black linear fits, respectively). (E) Cumulative number of vesicles released for open-geometry conditions. Release during the 100 Hz train simulation with one release site (solid red line) is approximately half the diffusion-mediated AZ supply rate for the open-geometry configuration (dashed red line, and solid back line fit). (F,G)

*Figure 8 continued on next page*

*Figure 8 continued*

Supply rate between 50–100 s and pool size computed from linear fits to data in **D** and **E**. Dark gray lines in **F** as for **A**. Light gray shaded regions denote range of experimentally measured values.

The following figure supplement is available for figure 8:

**Figure supplement 1.** Cartoon of the three vesicle pools and their measured transition rates for a single MFT AZ.

supply to the AZ. These AZ simulations predict the size of the experimentally measured RP and the vesicle replenishment rate from the reserve pool. Hence, our results identify the major physical determinants of vesicle diffusion within the crowded environment of presynaptic terminals, show vesicle mobility at an excitatory central synapse is comparable to that at ribbon synapses and suggest passive diffusion limits vesicle supply to AZs during sustained high-frequency release. Moreover, our results provide a structural basis for the functionally defined ~300 vesicle RP at MFT AZs.

## Physical determinants of vesicle mobility within MFTs

Our results show that crowding within presynaptic terminals gives rise to two types of physical interaction that slow vesicle mobility. Fast hydrodynamic interactions arising from fluid displacement influence vesicle mobility on both short (microsecond) and long (millisecond to second) timescales. In contrast, effects of vesicle collisions (*Gaffield et al., 2006*) are only felt on long time scales. The combined effects of these interactions result in a vesicle diffusion coefficient that evolves with a characteristic time course that depends on the vesicle density (*Figure 5C*). Given the strength of hydrodynamic interactions in crowded environments, it is surprising their effects have only recently been investigated in biological systems in the context of protein diffusion (*Doster and Longeville, 2007*; *Ando and Skolnick, 2010*). Our results extend these studies by showing hydrodynamic interactions are likely to be a major determinant of vesicle diffusion in synaptic terminals. Indeed, taking account of hydrodynamic and steric interactions arising from vesicle crowding explains much of the discrepancy between $D_{long}$ of vesicles and 50 nm beads in cytoplasm (*Luby-Phelps et al., 1987*). Hence, our results identify the main physical determinants of vesicle mobility in a central axon terminal and highlight the need to consider hydrodynamic interactions within crowded intracellular environments.

## Comparison of vesicle mobility across synapses

By quantifying how hydrodynamic and steric interactions vary with vesicle volume fraction and the fraction of immobile vesicles, it is possible to predict the vesicle mobility near the AZ and across different types of synapses (*Table 2*). At MFTs our quantitative model of vesicle mobility predicts that vesicle diffusion slows down from 0.025 $\mu m^2$/s within the MFT interior to 0.012 $\mu m^2$/s in the vicinity of the AZ. Thus, physical interactions slow vesicle mobility even in the absence of binding to tethers and connectors (which could slow diffusion further to 0.002 $\mu m^2$/s). For ribbon-type bipolar cells, which have a vesicle density comparable to MFTs, our model of vesicle mobility predicts $D_{long}$ = 0.014 $\mu m^2$/s, which matches well to the value measured in goldfish (0.015 $\mu m^2$/s; *Holt et al., 2004*), but cannot account for the exceptionally high mobility found in lizard (*Rea et al., 2004*). Nevertheless, our framework does predict the low vesicle mobility at the NMJ (predicted 0.006 $\mu m^2$/s vs. measured 0.005 $\mu m^2$/s; *Gaffield and Betz, 2007*), where vesicle density is more than double that within the MFT (0.33 vs. 0.17 volume fraction). Measurements of vesicle mobility in small central synapses are more numerous but highly variable. Our modelling predicts that $D_{long}$ is low within the middle of the vesicle cluster in small boutons (0.002 $\mu m^2$/s). This is similar to some FRAP-based estimates (*Shtrahman et al., 2005*) and recent single-vesicle tracking measurements (0.003 $\mu m^2$/s; *Lee et al. 2012*). Moreover, our predicted $D_{long}$ falls within the wide range of estimates of vesicle mobility using fluorescence correlation spectroscopy (*Figure 2D*, pink symbols; *Yeung et al., 2007*; *Jordan et al., 2005*), but these estimates depend on the diffusion/binding model applied (*Jordan et al., 2005*; *Shtrahman et al., 2005*; *Yeung et al., 2007*). Synapsin-based immobilization of vesicles appears prevalent in small boutons (*Orenbuch et al., 2012*), consistent with reports that most vesicles are immobile (*Shtrahman et al., 2005*; *Lee et al., 2012*). Moreover, there is a rapid vesicle exchange with the axon (*Staras et al., 2010*; *Herzog et al., 2011*) where mobility is high

**Table 2.** Predictions of vesicle mobility for different types of synaptic terminals.

| | MFT | MFT | MFT | NMJ | Boutons | Ribbon |
|---|---|---|---|---|---|---|
| | Centre | Cloud | AZ face | AZ cluster | AZ cluster | Centre |
| Ves. density (per $\mu m^2$) | 118 | 103 | 170 | 224 | 200 | |
| Ves. density (per $\mu m^3$) | 3930 | 3444 | 5652 | | | 4421 |
| Total ves. volume % | 17 | 17 | 25 | 33 | 29 | 29 |
| Immobile vesicle % | 25 | 25 | 17 | 40 | 73 | 13 |
| Imm. ves. volume % | 4 | 4 | 4 | 13 | 20 | 4 |
| Non-diffusible vol. % | 28 | 36 | 29 | 0 | 0 | 0 |
| $D_{cyto} / D_0$ | 0.01 | 0.01 | 0.01 | 0.01 | 0.01 | 0.01 |
| $D_{short} / D_{cyto}$ | 0.47 | 0.47 | 0.39 | 0.24 | 0.19 | 0.37 |
| $D_{long} / D_{short}$ | 0.41 | 0.30 | 0.29 | 0.20 | 0.09 | 0.43 |
| $D_{long} / D_{cyto}$ | 0.19 | 0.14 | 0.11 | 0.05 | 0.02 | 0.16 |
| $D_0$ ($\mu m^2/s$) | 12.682 | 12.682 | 12.682 | 12.764 | 12.764 | 9.055 |
| $D_{cyto}$ | 0.127 | 0.127 | 0.127 | 0.128 | 0.128 | 0.091 |
| $D_{short}$ | 0.060 | 0.060 | 0.050 | 0.031 | 0.025 | 0.033 |
| AZ wall hydro. ($\beta$) | | | 0.84 | | | |
| $D_{long}$ | **0.025** | **0.018** | **0.012** | **0.006** | **0.002** | **0.014** |
| $D_{long}$ measured | 0.025 | NA | NA | 0.005 | 0.004 | 0.110 (0.015) |

For the MFT, 3D vesicle (ves) densities were computed from 2D densities by diving by the slice thickness (0.03 µm). The total vesicle volume % was computed assuming a 44 nm vesicle diameter in fixed tissue. It was assumed the immobile vesicle volume fraction near the AZ was the same as in the MFT centre (4%). For the MFT centre computation, the non-diffusible volume (vol) is the mitochondria volume fraction; for the cloud and AZ face computations, the non-diffusible volume is the non-diffusible space within the vesicle clouds computed from the 14 AZ reconstructions. $D_0$ was computed via the Stokes-Einstein equation assuming a 49 nm vesicle diameter for in vitro conditions at 35°C. Diffusion constants and ratios are from Results (see **Figure 5**). Hydrodynamic (hydro) effects from the membrane wall near the AZ were computed via **Equations (9) and (10)**, and are average $\beta$ between 50 and 100 nm from the wall, where $\beta = (2\beta_{\parallel}+\beta_{\perp})/3$. Measured $D_{long}$ near the AZ face is not available (NA): vesicles close to AZs are too small to be detected by our FRAP measurements.

For the NMJ, the 2D vesicle density is the average of those reported in **Mantilla et al. (2004)** and **Coleman et al. (2008)**. The total vesicle volume % was computed assuming a proportional relationship with the MFT vesicle density and volume fraction. The immobile vesicle % and measured $D_{long}$ is from **Gaffield and Betz (2007)**. $D_0$ was computed assuming a 49 nm vesicle diameter and 37°C.

Data for the ribbon synapse is from **Rea et al. (2004)**. The 3D vesicle density was computed assuming 250,000 vesicles with 50 nm diameter inside a hemisphere with 6 µm diameter. The non-diffusible volume was set to zero since **Figure 3A** of Rea et al. shows few mitochondria. $D_0$ was computed assuming a 50 nm vesicle diameter and 22°C. Note, our estimate of $D_{long}$ is 10-fold smaller than the measured $D_{long}$ of Rea et al., but is comparable to the measured $D_{long}$ of another study of ribbon-type synapses in bipolar cells (value shown in brackets; 0.015 µm²/s; **Holt et al., 2004**).

For the hippocampal boutons, the 2D vesicle density is from **Li et al. (1995)** and **Schikorski and Stevens (2001)**. The total vesicle volume % was computed assuming a proportional relationship with the MFT vesicle density and volume fraction. The immobile vesicle % is from **Shtrahman et al. (2005)**. $D_0$ was computed assuming a 49 nm vesicle diameter and 37°C. Values for measured $D_{long}$ derived from fluorescence correlation spectroscopy (FCS) vary widely, depending on the model used to fit to the data ($5 \times 10^{-5}$ to 0.054 µm²/s; **Figure 2D**, pink symbols), but our predicted $D_{long}$ most closely matches that of a fit to pure diffusion (0.0043 µm²/s; **Shtrahman et al., 2005**) and the measured $D_{long}$ of Lee et al. (0.003 µm²/s; 2012) who tracked single vesicles using quantum dots.

For all terminals, $D_{cyto}/D_0$ was assumed to equal that in the MFT centre (0.01). $D_{short}/D_{cyto}$ was computed as the blue line in **Figure 5B**. $D_{long}/D_{short}$ was computed using data in **Figure 5A**.

(**Lee et al., 2012**). Our simulations suggest that the low mobility of vesicles in small central synapses is largely due to strong steric and hydrodynamic interactions arising from the high vesicle density. Moreover, although the presence of tethers/connectors will restrict vesicle mobility further, their effects will only be strong if they have long lifetimes (i.e. slow unbinding). While the properties of vesicle mobility within small synapses is still uncertain, our results show that to understand the variations in vesicle mobility across synapses, it is necessary to account for the physical interactions arising from organelle crowding.

## Supply of vesicles to the AZ and structural correlates of the releasable pool

Our 3D EM reconstructions revealed the AZs of MFTs are small and surrounded by a cloud of vesicles that is highly variable in shape and size. Our analysis shows that on average there are only 2 vesicles at the AZ face, suggesting few vesicles are docked and release ready at each AZ. This potentially explains why multi-vesicular release (*Wadiche and Jahr, 2001*) is not a dominant form of release at MFTs (*Sargent et al., 2005*). Away from the AZ face, the vesicle density falls to a level comparable to the density measured at the centre of the MFT. The accumulation of vesicles at the AZ face has a similar spatial extent as the AZ cytomatix protein Bassoon (~100 nm), which speeds vesicle reloading at MFTs (*Hallermann et al., 2010*) possibly via vesicle clustering (*Mukherjee et al., 2010*). Our experimentally constrained simulations suggest diffusion can supply vesicles to the AZ at the onset of high-frequency signalling (<100 ms) faster than the measured vesicle reloading rates of 40–80 s$^{-1}$ (*Figure 7C*), even in the presence of protein filament connectors and tethers (*Siksou et al., 2007*; *Fernández-Busnadiego et al., 2013*). This suggests factors other than diffusion, such as docking and priming, limit vesicle reloading at the onset of sustained release. Ca$^{2+}$-dependent mechanisms that speed docking and priming (*Neher and Sakaba, 2008*) would therefore be most effective at accelerating vesicle reloading at the early stages of high-frequency signalling, but increasing vesicle mobility or reducing the immobile fraction would be effective later, once vesicle depletion occurs.

Previous studies of vesicular release at cerebellar MFTs (*Saviane and Silver, 2006*; *Hallermann et al., 2010*; *Sargent et al., 2005*) and vestibular nerve synapses (*McElvain et al., 2015*) suggest that each AZ has an RRP of 1–2 vesicles and a RP of ~200–300 vesicles (*Figure 8—figure supplement 1*). Remarkably, 3D AZ reconstructions predict the size of the RRP and our simulations predict the RP from the spatial positions of the vesicles, the diffusible space surrounding the AZ and our model of vesicle mobility. Furthermore, once the RP is depleted, our simulations predict a vesicle supply rate of 7 s$^{-1}$, which matches the release rates of 7–8 s$^{-1}$ recorded at these synapses (*Saviane and Silver, 2006*; *McElvain et al., 2015*). Because glutamate refilling is 100-fold slower than these rates (*Hori and Takahashi, 2012*), rapid local endocytosis (*Watanabe et al., 2013*) is unlikely to be responsible for these limiting rates. Our results therefore suggest that, during sustained high-frequency release, vesicle supply from the reserve pool to the AZ is limited by vesicle diffusion from the interior of the MFT. Thus, diffusion rather than docking and priming potentially limits the rate at which continuous sensory variables, such as head velocity (*Arenz et al., 2008*) and joint angle (*van Kan et al., 1993*), are transmitted. Moreover, the fact that increasing diffusional access to the AZ effectively eliminates the functionally defined RP indicates the shape and extent of the vesicle cloud associated with an AZ is a major determinant of the RP, because it sets the rate of supply from the reserve pool. This structural property could therefore contribute to the heterogeneous functional properties of MFTs (*Sargent et al., 2005*) originating from different precerebellar nuclei (*Chabrol et al., 2015*). Thus, our results provide the structural basis for a functionally defined vesicle pool and show diffusion ultimately limits vesicle supply during sustained high-frequency signalling at a central synapse.

## Materials and methods

### Animals

The generation and general characterization of the VGLUT1$^{Venus}$ (Slc17A7$^{ct(venus)Nbr}$) knock-in mouse line was published previously (*Herzog et al., 2011*); all experiments using this line were performed with littermates derived from crossing homozygous VGLUT1$^{v/v}$ mice (F2 SV129/ola x C57BL/6 genetic background). All animal experiments were conducted in strict accordance with the United Kingdom Home Office Animals Scientific Procedures Act of 1986, and approved by the UCL ethics review board. All mice were anaesthetized with ketamine or isoflurane during surgical procedures.

### Electron microscopy and 3D reconstructions

Two C57Bl6 mice (P28 and P30) were anaesthetized with ketamine (35 mg) and transcardially perfused with 0.9% saline, then with 2% paraformaldehyde and 1% glutaraldehyde in 0.1 M Na-acetate buffer (pH = 6) for 2 min, then with 2% paraformaldehyde and 1% glutaraldehyde in 0.1 M Na-

borate buffer (pH = 8) for one hour. Four VGLUT1[Venus] mice (all P28) were anaesthetized with keta-mine (35 mg) and transcardially perfused with 0.9% saline, then with 4% paraformaldehyde and 0.1% or 0.05% glutaraldehyde in 0.1 M Na-phosphate buffer for 25 min. After perfusion, brains were dis-sected and 60 μm sections were cut from the cerebellar vermis. Sections from VGLUT1[Venus] mice were immunoreacted for either VGLUT1 or GFP using anti-VGLUT1 (1:500 dilution; Synaptic Systems Cat# 135 302, RRID:AB_887877; Goettingen, Germany) or anti-GFP (1:1000 dilution; Millipore Cat# AB3080P, RRID:AB_91338; Billerica, Massachusetts) primary antibodies, respectively. Sections were then washed, incubated with biotinylated or 0.8 nm gold-coupled (Aurion, Wageningen, The Nether-lands) secondary antibodies. Reactions were visualized with either silver enhancement (Aurion SE-EM kit) or a DAB reaction. Following reactions, sections from the C57Bl6 mice were washed in 0.1 M PB then treated with 1% $OsO_4$ and 1% uranyl acetate before dehydration and embedding in Epoxy resin. Small blocks from the sections were re-embedded and 30 nm serial ultrathin sections were cut for 3D reconstruction (*Holderith et al., 2012*). Images were taken with a Jeol JEM1011 electron microscope equipped with a bottom-mounted CCD camera (Cantega; Olympus Soft Imaging Solu-tions, Münster, Germany). 3D reconstructions and measurements were performed using Synapse Web Reconstruct (RRID:SCR_002716; http://synapses.clm.utexas.edu/tools/reconstruct/reconstruct. stm). MFTs from VGLUT1[Venus] mice were qualitatively analysed at low and high magnification and compared to those obtained from C57Bl6 mice. All GFP- and VGLUT1-immunopositive MFTs were large, contained clusters of mitochondria and had vesicle densities similar to those found in C57Bl6 mice. The cloud of vesicles surrounding the AZs were also apparent with variable size. The proximity of mitochondria to AZs was also highly variable, similar to that found in C57Bl6 mice (data not shown).

## Electron tomography

Serial sections (200 nm) from C57Bl6 mice were cut and collected onto copper slot grids. Fiducial markers were introduced at both sides of the grids (*Imig et al., 2014*). Single-axis tilt series were acquired in FEI Tecnai G2 Spirit BioTWIN transmission EM operating at 120 kV and equipped with an Eagle 4K HS digital camera (FEI, Eindhoven, The Netherlands). Tomographic volumes were recon-structed using IMOD (*Kremer et al., 1996*; RRID:SCR_003297) and exported as z-stacks for analysis. Vesicle diameters were measured using Reconstruct, defined as the distance between the outer parts of the membrane bilayers at the plane where the diameter was largest.

## Quantification of vesicle count and density surrounding the AZ

Vesicle count was computed as a function of distance from the AZ face for 14 3D reconstructions. To do this, the distance from a vesicle's centre to the nearest point of the AZ face was measured for each vesicle and a histogram of the distances constructed using 22 nm bins. In order to compute the vesicle density as a function of distance to the AZ, we limited the diffusible space to the outermost extremity of the vesicle 'cloud'. To do this, we divided the space surrounding the AZ into 44 nm vox-els and defined voxels as diffusible only if they contained any part of a vesicle (*Figure 7A*). The vesi-cle density was computed within the cloud as a function of distance from the AZ by sorting the diffusible voxels as a function of their distance to the AZ and counting the number of vesicle centre points that fell within each bin and dividing by the sum of the voxel volumes. The vesicle density was also calculated by subdividing the 44 nm voxels into 4.4 nm voxels and computing the volume of voxels that fell within the vesicles and dividing by the total volume of the diffusible voxels. To calcu-late the vesicle density when assuming all space surrounding the AZ is diffusible, we divided the ves-icle count as a function of distance from the AZ (22 nm bins, as described above) by the space surrounding the AZ in 22 nm thick bands.

## FRAP recordings

Parasagittal slices of the cerebellar vermis were prepared (*Nielsen et al., 2004*) from VGLUT1[Venus] knock-in mice (P22–33, n = 19). FRAP in the cerebellar slices was performed on a custom spot confo-cal system (Prairie Technologies, Middleton, Wisconsin; *DiGregorio et al., 2007*) at 35°C unless stated otherwise. Data was acquired and analyzed using NeuroMatic (RRID:SCR_004186; http:// www.neuromatic.thinkrandom.com) that runs within the IGOR Pro environment (RRID:SCR_000325; WaveMetrics, Portland, Oregon). Laser light (488 nm) was focussed to a diffraction-limited spot with

100x objective lens (1.0 NA, Olympus). The bleaching pulse was 28.5 µW (after the objective) of 0.5 ms duration and probe pulses were 0.04 µW of 2 ms duration. The ACSF contained (in mM) 125 NaCl, 2.5 KCl, 2 CaCl$_2$, 1 MgCl$_2$, 1.25 NaH$_2$PO$_4$, 26 NaHCO$_3$ and 25 glucose. Okadaic acid (2 µM), cytochalasin-D (10 µM) plus latrunculin-B (10 µM), jasplakinolide (2 or 5 µM) and roscovitine (50 µM) were added to the ACSF where specified. FRAP recordings were discarded if they had an unstable baseline fluorescence or large jumps in fluorescence. MFTs were discarded if they had less than 3 FRAP recordings.

To test for potential phototoxicity, we repeated our measurements using half the laser power during the bleaching pulse (16 MFTs, 156 total locations, 1 mouse), which produced ~13% bleaching from baseline fluorescence, rather than ~35% under control conditions, and computed $f_{1s}$ and $f_{5s}$ values. However, no significant differences were found for half-power conditions compared to control conditions for $f_{1s}$ (40 ± 5 vs. 35 ± 2%, respectively, p=0.4, unequal-variance $t$-test) and $f_{5s}$ values (59 ± 7 vs. 63 ± 2%, p=0.6, unequal-variance $t$-test), consistent with results of finite-difference FRAP simulations for full- and half-power bleaching (not shown). We also tested for time-dependent changes in the FRAP recordings within single MFTs by comparing $f_{1s}$ and $f_{5s}$ distributions computed from the first 4–10 and last 4–10 recordings taken from a single MFT. Restricting the analysis to those MFTs with at least 8 recordings (n = 47, 6 mice), we found no difference between $f_{1s}$ values of the first and last recordings (38 ± 2 and 37 ± 2%, respectively, p=0.8, paired $t$-test) and $f_{5s}$ values of the first and last recordings (63 ± 3 and 63 ± 3%, p=0.8, paired $t$-test) indicating the laser was not inducing time-dependent changes in vesicle mobility within single MFTs.

To test for photoactivation of bleached Venus (*McAnaney et al., 2005*), we compared FRAP recordings with 21 probe pulses after the large bleaching pulse (n = 82; *Figure 1B*, top) to recordings with 11 probe pulses (n = 79; recorded within the same MFTs from a P45 and P51 mouse) and found no significant difference ($f_{1s}$ = 33 ± 5 vs. 41 ± 4%, respectively, p=0.17, unequal-variance $t$-test). Since photoactivation of bleached Venus should result in a faster fluorescence recovery with a larger number of probe pulses, these results demonstrate the small brief probe pulses used in our FRAP experiments did not cause photoactivation of bleached Venus.

For each population of FRAP recordings from multiple MFTs, a final weighted average ($X_i$) and variance ($\sigma_i^2$) for a given probe pulse $i$ was computed across MFTs (e.g. *Figure 1D*; black circles) using the following equations:

$$\begin{aligned}
\bar{X}_i &= \sum_j n_j \bar{x}_{ji}/N_1 \\
\sigma_i^2 &= \sum_j n_j (\bar{x}_{ji} - \bar{X}_i)^2\, N_1/(N_1^2 - N_2) \\
N_1 &= \sum_j n_j \quad N_2 = \sum_j n_j^2
\end{aligned}$$

(1)

where $j$ is the MFT index number, $n_j$ the number of recordings for a given MFT, and $x_{ji}$ the average of these $n_j$ recordings.

## Estimation of D$_{long}$ from FRAP

A theoretical fluorescence recovery curve for pure diffusion can be described as follows (*Axelrod et al., 1976*):

$$F_K(t) = f_0 \sum_n \left[\frac{-K^n}{n!}\right]\left[1 + n\left(1 + \frac{2t}{\tau_D}\right)\right]^{-1}$$

(2)

where $f_0$ is the fluorescence before bleaching, $K$ is the bleaching parameter that determines the level of fluorescence immediately after bleaching ($f_b$ = (1−exp(−K))/K), and $\tau_D$ is the characteristic diffusion time defined as $\tau_D = \omega^2/4D_{long}$. Parameter $\omega$ is the half-width of the illumination beam at e$^{-2}$ of the peak height which we estimated to be 0.23 µm from our iPSF. We computed the $F_K$ summation for $n$ = 0–19, which was sufficient to approximate the infinite series. To determine D$_{long}$, *Equation (2)* was fitted to our normalized control data by letting parameters $K$ and D$_{long}$ vary while fixing $f_0$ = 1 and $\omega$ = 0.23 µm. To allow a variable steady-state fluorescence ($f_{inf}$) during the fit, $F_K$ was transformed as follows: $f_K = f_b + (f_{inf} - f_b)(F_K - f_b)/(1 - f_b)$. Results of the fit were: D$_{long}$ = 0.018 ±

0.005 $\mu m^2$/s, $K$ = 0.94 ± 0.05, $f_{inf}$ = 0.917 ± 0.011. After drift correction, results of the fit were: $D_{long}$ = 0.025 ± 0.003 $\mu m^2$/s, $K$ = 0.97 ± 0.03, $f_{inf}$ = 0.883 ± 0.004.

One caveat of using *Equation (2)* to estimate $D_{long}$ is that its accuracy relies on the following assumptions about our experimental paradigm: (1) the bleaching pulse is brief compared to the rate of diffusion, (2) bleaching is a simple irreversible first-order reaction, (3) the bleaching volume is small compared to the total volume of the synaptic terminal (4) iPSF and cPSF have the same Gaussian intensity profile in the *xy* direction and are infinite in the *z* direction (i.e. both are the same Gaussian beam), (5) fluorescence detection causes no additional bleaching during the fluorescence recovery phase, (6) there is a homogeneous concentration of diffusant and (7) diffusion is isotropic. While assumptions (1), (2) and (3) are reasonable for our experimental paradigm, assumptions (4–7) may not be strictly correct. To test assumptions (4) and (5), we used a 3D finite-difference reaction-diffusion simulation approach (see below) that explicitly modelled the bleaching and probe pulses and the spatial properties of the iPSF and cPSF of our microscope (*Figure 3—figure supplement 2*). Comparison of the fit to our drift-corrected control data (where $D_{long}$ = 0.025 $\mu m^2$/s) to a finite-difference simulation with $D_{long}$ = 0.025 $\mu m^2$/s showed a close agreement, with only a slightly slower rate of recovery for the finite-difference simulation (not shown; $t_{1/2}$ = 0.58 vs. 0.66 s). Only a small increase in $D_{long}$ to 0.028 $\mu m^2$/s of the finite-difference simulation was necessary to produce matching FRAP curves (*Figure 2C*). Similarly, to test assumptions (4–7), we used a 3D Monte Carlo simulation approach (see below) that included long cylindrical mitochondria, which form non-diffusible regions and are therefore likely to introduce anisotropic diffusion. Nevertheless, we found a close agreement to the finite-difference simulation that simulates a homogeneous concentration with isotropic diffusion (*Figure 4D*). These results suggest that the assumptions required to apply *Equation (2)* are reasonable for our experimental paradigm and it should therefore produce a reasonable estimate of $D_{long}$.

## Finite-difference simulations

FRAP experiments were simulated in 3D space using an extended version of D3D, an in-house finite-difference reaction-diffusion simulator (*Nielsen et al., 2004*; *DiGregorio et al., 2007*; *Nakamura et al., 2015*). The voxel size was 50 nm and the time step (*dt*) was set by a stability restriction parameter (*Crank, 1975*) which was tested to be sufficiently small. The expression for iPSF was the same as that derived for the measured emission PSF of our microscope (*Figure 3—figure supplement 2A,B*), but using a light wavelength of 488 nm instead of 515 nm. The expression of cPSF (a Gaussian function with $FWHM_{xy}$ = 255 nm and $FWHM_z$ = 916 nm) was derived from fluorescence measured from 110 nm beads (*Figure 3—figure supplement 2C*). Both iPSF and cPSF were positioned so their peaks were aligned at the centre of the simulation geometry (*Figure 3C*). Values of iPSF and cPSF were computed at the centre of each voxel, and these values were used for bleaching and fluorescence detection. The $e^{-2}$ volume of iPSF (0.31 $\mu m^3$) was computed by summing the volume of voxels with an iPSF value $> e^{-2}$. The $e^{-2}$ volume of the cPSF was 0.15 $\mu m^3$, which is small compared to the volume of the MFTs (30–60 $\mu m^3$; *Jakab and Hámori, 1988*; *Kim et al., 2013*). To take advantage of *xy* symmetry and reduce simulation time, we simulated one-quarter of the entire space.

## Monte Carlo simulations

Brownian motion of vesicles, including steric interactions, was simulated using a 3D Monte Carlo algorithm for non-overlapping hard spheres (*Cichocki and Hinsen, 1990*). At the start of the simulation, mobile vesicles were given the same $D_{short}$, the mean vesicle step size (*dr*) was set to a small fraction (0.5–5.0 nm) of the vesicle diameter (49 nm), and *dt* was computed via Einstein's relation in 3D space:

$$dt = dr^2/6D_{short} \tag{3}$$

If a displaced vesicle resulted in overlap with another vesicle, or a non-diffusible voxel, the vesicle stayed at the same location; otherwise the vesicle was moved to the new location. Vesicles were not allowed to overlap with the simulation borders, except when computing $D(t)$, in which case periodic boundary conditions were used.

For FRAP simulations, mitochondria were simulated as cylindrical regions of non-diffusible voxels with 0.28 µm diameter and 2.25 µm length, randomly placed throughout the cubic simulation space in clusters of 2–3 (*Figure 3C*). Vesicles were given an initial fluorescence (*f*) of unitary and this was then scaled by the following reaction during the bleaching and probe pulses:

$$f_{t+dt} = f_t[1.0 - iPSF(x,y,z) \cdot k \cdot dt] \tag{4}$$

where *t* is time, *dt* is the time step and *k* is the bleaching rate that is scaled by a normalized 3D spatial weighting function defined by iPSF described above. Average fluorescence of all vesicles was computed according to a normalized 3D spatial weighting defined by cPSF described above. The vesicle step size *dr* was set to 2 nm which was small enough to avoid discretization error (*Figure 4— figure supplement 1A*). We compared simulated FRAP curves for cubic geometries in the range 1.5–3.0 µm and found a 2 µm cube produced negligible border effects for simulations with $D_{short} < 0.080$ µm$^2$/s and was therefore sufficiently large to simulate the centre of a large MFT (*Figure 4— figure supplement 1B*).

For FRAP simulations with added drift (*Figure 2—figure supplement 1B*), mitochondria were simulated as non-diffusible volumes specified by *x*, *y* and *z* coordinates, rather than non-diffusible voxels as shown in *Figure 3C*, so that drift could be applied to the mitochondria's *x*, *y* and *z* coordinates. If a vesicle moved outside the geometry it was returned to the opposite side of the geometry (i.e. periodic boundary conditions) with a fluorescence value of 1.0 to simulate unbleached vesicles moving into the simulation space.

One potential caveat to our quantification of vesicle mobility is the possibility our measured FRAP represents the movement of small clumps of vesicles rather than individual vesicles. However, this should not unduly affect our results since the vesicle volume fraction, rather than the vesicle size, primarily determines the hydrodynamic and steric contributions to $D_{long}$ (*Medina-Noyola, 1988*). Moreover, inspection of EMs of the interior of the MFT suggests vesicles are dispersed. Interestingly, vesicles exhibit a negative potential, which generates an electrostatic repulsive force, which could explain why vesicles do not tend to aggregate in clumps (*Ohsawa et al., 1981*).

For AZ simulations, we used the 3D reconstructions from our EM data. The diffusible space was determined by the outer surface of the vesicle cloud as described above. In one of the 14 AZ reconstructions, a low vesicle density resulted in space close to the AZ being 'filled in' by our algorithm. As the presence of non-diffusible space so close to the AZ seemed unlikely, we expanded the diffusible space surrounding this AZ by two vesicle diameters (the dimensions of the high-vesicle-density zone; *Figure 6C*). To remove the effects of fixation shrinkage in our EM data, the geometries, including AZs, were scaled up by 11% (*Korogod et al., 2015*). After scaling, vesicle diameters and voxel widths were 49 nm.

Because we used a uniform vesicle diameter of 49 nm, rather than a distribution, small overlaps between vesicles occurred. To alleviate this problem we used an algorithm (*Lubachevsky and Stillinger, 1990*) that first shrank vesicles until there were no overlaps and then allowed them to randomly move, slowly expanding until they reached a diameter of 49 nm. To maintain high vesicle densities close to the AZ (*Figure 6C*) vesicles <200 nm from the AZ were not allowed to move more than 24.5 nm from their original location; all other vesicles were not allowed to move more than 49 nm. We verified the average vesicle density near the AZ was similar before and after removing vesicle overlaps (*Figure 7—figure supplement 1A*). To create a vesicle reserve surrounding the reconstruction geometries, the geometries were expanded and populated with vesicles at a 17% volume fraction of which 25% were immobile (*Figure 7A*). Final simulations had ~13,000 mobile vesicles. To create 'open' geometries, non-diffusible voxels surrounding the AZ were converted to diffusible space and populated with vesicles as for the reserve. The final simulations had ~17,000 mobile vesicles. For all AZ simulations, *dr* = 5 nm, which was sufficiently small to avoid significant discretization error (*Figure 7—figure supplement 3A,B*). Simulations were repeated 20 times for each AZ and averaged.

To investigate how filament connectors might influence vesicle properties near the AZ, we performed simulations whereby vesicles <150 nm from the AZ rapidly formed connectors with neighbouring vesicles (average 1.5 connectors per vesicle; *Siksou et al., 2007*; *Fernández-Busnadiego et al., 2013*) if they were <10 nm of one another. Once connected, vesicles were not allowed to diffuse more than 10 nm from each other, thereby simulating flexible protein filaments (*Graydon et al., 2014*). A connector on rate of 10,000 s$^{-1}$ was used and each bound connector had

a lifetime that was randomly sampled from an exponential distribution with mean equal to the inverse of the connector unbinding rate (1000 or 10 s$^{-1}$). Filament tethers to the AZ (*Fernández-Busnadiego et al., 2013*) were simulated in a similar fashion. In this case, vesicles became 'tethered' to the AZ if they were <8 nm from it. The same on and off rates of the connectors were used and no more than 2 vesicles could be tethered to the AZ at one time.

Because the vesicle density increases close to the AZ (*Figure 6C*) the effects of vesicle-vesicle hydrodynamic interactions are expected to be larger near the AZ. Hence, for those vesicles within the vesicle cloud, we used the local vesicle volume fraction surrounding each vesicle to compute the vesicle's $D_{short}$ and *dr* on every time step of the AZ simulations. The local vesicle density was computed within a distance of 4 vesicle radii from the vesicle's centre (*Urbina-Villalba et al., 2003*). $D_{short}$ was computed from the ratio $D_{short}/D_{cyto}$ (*Figure 5B*) assuming our estimate of $D_{cyto}$ = 0.127 $\mu m^2$/s and 0.04 immobile vesicle volume fraction.

To compute the vesicle supply rates in *Figure 7*, release events were counted within the following windows: 0-10 ms in 2 ms bins, 10-100 ms in 10 ms bins, 100-1000 ms in 100 ms bins, 1-10 s in 1 s bins, 10-100 s in 10 s bins. Release counts were converted to rates by dividing the bin count by the bin duration and the number of simulation repetitions (20). The cumulative vesicle counts were computed within the same windows. The vesicle release rates and cumulative vesicle counts in *Figure 8* were computed in the same manner, except for using a single 10 ms bin within the first window as this corresponds to the stimulus interval of the 100 Hz train. Final release rates and counts were displayed as continuous functions by drawing lines between the midpoint of each consecutive bin.

## Calculation of hydrodynamic interactions

Effects of hydrodynamic interactions on vesicle mobility from vesicle-vesicle interactions were determined with analytical expressions for $D_{short}/D_{cyto}$ as a function of the vesicle volume fraction ($\Phi$). For conditions of all mobile vesicles, $D_{short}/D_{cyto}$ (here denoted as $\Gamma_m$) was computed using the analytical expression of *Tokuyama and Oppenheim (1994)* (*Figure 5B*; red line):

$$\frac{D_{short}}{D_{cyto}} = \Gamma_m = \frac{1}{1+H(\phi_m)}$$
$$H(\phi_m) = \frac{2b^2}{1-b} - \frac{c}{1+2c} - \frac{bc(2+c)}{(1+c)(1-b+c)}$$
$$b(\phi_m) = \sqrt{9\phi_m/8}$$
$$c(\phi_m) = 11\phi_m/16$$

(5)

where $D_{cyto}$ is equivalent to $D_0$ of Tokuyama and Oppenheim and $\Phi_m$ is the mobile vesicle volume fraction. For a mixture of mobile and immobile vesicles, $D_{short}/D_{cyto}$ (here denoted as $\Gamma_{mix}$) was computed using the self-consistent equation of *Freed and Muthukumar (1978)* up to the squared term:

$$\zeta_{short} = \zeta_{cyto}\left(1 + \kappa r + \frac{(\kappa r)^2}{3}\right)$$
$$\zeta_{cyto} = 6\pi\eta r$$
$$\kappa = \sqrt{c_{im}\zeta_{short}/\eta}$$
$$\frac{D_{short}}{D_{cyto}} = \Gamma_{mix} = \frac{\zeta_{cyto}}{\zeta_{short}}$$

(6)

where $\zeta$ is the drag coefficient (related to Stokes-Einstein equation $D = k_BT/\zeta$, $k_B$ being the Boltzmann's constant and T absolute temperature), $\eta$ is viscosity, $r$ is the vesicle radius and $c_{im}$ is the density of immobile vesicles (count per $\mu m^3$) which was converted to the immobile vesicle volume fraction ($\Phi_{im}$). However, *Equation (6)* describes conditions for small $\Phi_m$. For conditions with large $\Phi_m$, hydrodynamic interactions from the immobile vesicles are expected to be less due to 'screening' effects from the mobile vesicles. To account for this, we applied the analytical model of *Michailidou et al. (2009)* to compute $\Gamma_{mix}$ in the presence of a large $\Phi_m$ (*Figure 5B*; blue solid line) denoted as $\Gamma'_{mix}$:

$$\Gamma'_{mix} = \frac{\Gamma_m}{1+\frac{\Gamma_m}{\Gamma_{ff}}\left(\frac{1}{\Gamma_{mix}}-1\right)}$$
$$\Gamma_{ff} = 1 - 1.5\phi_m + 0.75\phi_m^2$$

(7)

where $\Gamma_{ff}$ is the far-field-only $D_{short}$, i.e. the short-time self-diffusion coefficient in the absence of

near-field hydrodynamic interactions, derived from previously published computer simulations (*Banchio and Brady, 2003*). Note that when $\Gamma_{ff} \approx \Gamma_m$ (e.g. with a low $\Phi_m$) then *Equation (7)* reduces to $\Gamma'_{mix} \approx \Gamma_m \Gamma_{mix}$.

The combined long-time effects of steric and hydrodynamic interactions, expressed as $D_{long}/D_{cyto}$, was computed for conditions of all mobile vesicles using the analytical expression of *Tokuyama and Oppenheim (1994)*:

$$\frac{D_{long}}{D_{cyto}} = \frac{1 - 9\phi_m/32}{1 + H(\phi_m) + (\phi_m/\phi_0)/(1 - \phi_m/\phi_0)^2}$$

$$\phi_0 = \frac{(4/3)^3}{7 ln(3) - 8 ln(2) + 2} \approx 0.5718 \tag{8}$$

where $\Phi_m$ is the mobile vesicle volume fraction and $H(\Phi_m)$ is defined in *Equation (5)*. This equation was used only for comparison purposes in *Figure 5D*.

For AZ simulations, the effects of hydrodynamic interactions from a membrane wall (denoted as $\beta$) were computed via analytical expressions for diffusion toward a wall ($\beta_\perp$; *Brenner, 1961*) and parallel to a wall ($\beta_\parallel$; *Goldman et al., 1967*):

$$\beta_\perp = \frac{6\Delta_z^2 + 2r\Delta_z}{6\Delta_z^2 + 9r\Delta_z + 2r^2}$$

$$\beta_\parallel = 1 - \frac{9}{16}\rho + \frac{1}{8}\rho^3 - \frac{45}{256}\rho^4 - \frac{1}{16}\rho^5 \tag{9}$$

$$\rho = r/\Delta_{z+r}$$

where $\Delta_z$ is the shortest distance between the wall (i.e. AZ) and edge of the vesicle and $\Delta_{z+r}$ is the distance between the wall (i.e. AZ) and the centre of the vesicle (i.e. $\Delta_{z+r} = \Delta_z + r$). To account for diminished hydrodynamic interactions from the wall due to a high $\Phi_m$, we again used the analytical model of *Michailidou et al. (2009)* to apply a correction factor to $\beta$, denoted as $\beta'$:

$$\beta'_\perp = \frac{1}{1 + \frac{\Gamma'_{mix}}{\Gamma_{ff}}\left(\frac{1}{\beta_\perp} - 1\right)}$$

$$\beta'_\parallel = \frac{1}{1 + \frac{\Gamma'_{mix}}{\Gamma_{ff}}\left(\frac{1}{\beta_\parallel} - 1\right)} \tag{10}$$

Values for $\beta'$ were then used to appropriately scale the vesicle steps in the *x*, *y* and *z* directions computed during each time step.

## Statistics

Data are presented as mean ± standard error of the mean (SEM) and fit parameters as ± standard deviation (STDV). Experimental and simulation results were compared with a chi-square criterion and experimental means were compared using the Student's *t*-test (unpaired two-tailed equal-variance unless stated otherwise) where $p < 0.05$ was considered significant. Model comparisons were computed via an F-test. No statistical method was used to predetermine sample sizes.

## Code

Java code to reproduce the finite-difference and Monte Carlo reaction-diffusion simulations is available at https://github.com/SilverLabUCL/D3D_eLife.

## Acknowledgements

We thank Nils Brose for supplying the VGLUT1[Venus] knock-in mice, Stefan Hallermann for providing electrophysiological data on wild-type mice, Noemi Holderith for performing electron tomography and Diccon Coyle for providing technical assistance. We would also like to thank J Ashmore, D Attwell, T Fernández-Alfonso, F Lanore, B Marin, A Valera and T Younts for their comments on the manuscript.

## Additional information

### Funding

| Funder | Grant reference number | Author |
|---|---|---|
| Agence Nationale de la Recherche | ANR-12-JSV4-0005-01VGLUT-IQ | Etienne Herzog |
| Agence Nationale de la Recherche | ANR-10-LABX-43BRAIN | Etienne Herzog |
| European Research Council | 293681 | Zoltan Nusser |
| Magyar Tudományos Akadémia | Momentum Grant, Lendület, LP2012-29 | Zoltan Nusser |
| European Commission | LSHM-CT-2005-019055 | Zoltan Nusser Robin Angus Silver |
| Wellcome Trust | 095667 | Robin Angus Silver |
| European Research Council | 294667 | Robin Angus Silver |
| Wellcome Trust | 203048 | Robin Angus Silver |

The funders had no role in study design, data collection and interpretation, or the decision to submit the work for publication.

### Author contributions

JSR, ZN, Conception and design, Acquisition of data, Analysis and interpretation of data, Drafting or revising the article; LK, Conception and design, Analysis and interpretation of data; EH, Developed transgenic mice, Drafting or revising the article; RAS, Conception and design, Analysis and interpretation of data, Drafting or revising the article

### Author ORCIDs

Jason Seth Rothman, http://orcid.org/0000-0003-3036-2291
Etienne Herzog, http://orcid.org/0000-0002-0058-6959
Zoltan Nusser, http://orcid.org/0000-0001-7004-4111
Robin Angus Silver, http://orcid.org/0000-0002-5480-6638

### Ethics

Animal experimentation: Animal experiments were conducted in strict accordance with the United Kingdom Home Office Animals Scientific Procedures Act of 1986, and approved by the UCL ethics review board. All mice were anaesthetized with ketamine or isoflurane during surgical procedures.

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
