## [Decision Letter]

Thank you for submitting your article "Physical determinants of vesicle mobility and supply at a central synapse" for consideration by *eLife*. Your article has been reviewed by two peer reviewers, one of whom is a member of our Board of Reviewing Editors, and the evaluation has been overseen by Gary Westbrook as the Senior Editor. The reviewers have opted to remain anonymous. The reviewers have discussed the reviews with one another and the Reviewing Editor has drafted this decision to help you prepare a revised submission.

Summary:

This manuscript examines vesicle mobility and supply at mossy fibre synaptic terminals (MFTs), to determine how these parameters contribute and are putatively rate limiting during high release activity. Using a combination of 3D ultrastructural analysis, measurements of vesicle diffusion and simulations, the authors convincingly conclude that vesicle diffusion is the main rate limiting factor in vesicle supply, which is surprising as this would limit the impact of vesicle tethering and priming processes at least on a long time scale, previously considered to be the critical rate limiting step in preparing vesicle for fusion.

Essential revisions:

A major criticism relates to the feasibility of using simple confocal FRAP measurements to obtain vesicle mobility values in this preparation:

1) As stated in the text MFTs in contrast to e.g. small hippocampal boutons offer the advantage of being clearly larger than the confocal bleaching volume. However, exactly this fact might account for the differences in measured SV mobilities compared to hippocampal boutons: the confocal volume for FRAP or FCS measurements in the latter case almost inevitably contains the volume close to the active zone (in axial direction the volume extends far beyond the small bouton boundaries). This is less likely the case in FRAP experiments in the MFTs which measured 7 to 10 µm according to the authors: thus, rather volumes in the middle of the MFT at some distance to the AZs where the SV density is lowest (Figure 6) are probed. Although multiple or at least three measurements were made per terminal at different locations the large axial extent of the bleaching volume implies that mostly volumes away from the small AZs are covered. Given this and that the '3D reconstructions revealed the AZs of MFTs are small and surrounded by a cloud of vesicles that is highly variable in shape and extent', could the measured fast rates of fluorescence recovery predominantly reflect fast SV movements at the fringes of these SV clouds and even in-between clusters from different AZs? Here, overall SV densities are small and thus the dynamics might be much higher than within the densely packed areas close to the AZ, where SVs in EM tomograms even appear interconnected by short filaments, which might highly reduce their mobility.

In summary, given the experimental design of the FRAP measurements and the geometry of MFTs with their inhomogeneous SV distributions, there was concern about the claim that these results 'show diffusion ultimately limits vesicle supply during sustained high-frequency signaling at a central synapse'! SV supply and mobility at the AZ simply have not been and cannot be easily measured in this preparation.

2) An important conclusion of the paper is that "the geometry of the diffusible space determines the size of the RPV", based on the fact that back extrapolation of the cumulative release provides a similar pool size as measured in MFTs. However, before arriving at this conclusion several aspects should be addressed/discussed:

a) Using linear back extrapolation is known to underestimate the RRP due to the fact that the replenishment rate is not constant but depends on the depletion of the RRP (for a full RRP at steady-state the net replenishment rate is 0) (Neher, 2015). This error is larger for higher replenishment rates, which could explain the low RRP for the open geometry. Since in the simulations all vesicles are tractable, accurate estimates of the replenishment rates over time should be calculated and used for more accurate RRP estimates in the default- and open geometry.

b) It is not clear how (molecular) priming is addressed in the model. Primed vesicles in the RRP are most likely docked at/tethered to the membrane via (partial?) assembly of SNARE-complexes and are therefore not free to diffuse (?). One would expect that this stabilizes an RRP of a certain size, independent of the geometry of the diffusible space but dependent on the number of available docking sites. This is supported by the fact that deletion of priming molecules like Munc13 and Munc18 produce a complete loss of RRP without affecting the geometry of the terminal. The impact of molecular priming on the RRP in the diffusion model should be discussed in the context of this evidence.

c) A well-established method to assess the RRP is the application of hypertonic solutions. Interestingly, although this manipulation most likely has a strong impact on the geometry of the diffusible space (shrinkage), concentrations beyond 500mM have a strong effect on release kinetics but not on the total release or steady-state priming rates at the end of the stimulus. The authors should discuss how these findings can be reconciled with the model.

3) Priming/RRP replenishment is sped up in a Ca^2+^ dependent manner during high frequency stimulation (Sakaba & Neher, 2001). This suggests an active flow of vesicles in the direction of the AZ during or a change in mobility properties of the vesicles. The authors should discuss how this fits in their model.

4) Software or scripts should be made available to the community as supplementary material via *eLife* or via a public repository for computational models for neuroscience.

---

## [Author Response]

*1) As stated in the text MFTs in contrast to e.g. small hippocampal boutons offer the advantage of being clearly larger than the confocal bleaching volume. However, exactly this fact might account for the differences in measured SV mobilities compared to hippocampal boutons: the confocal volume for FRAP or FCS measurements in the latter case almost inevitably contains the volume close to the active zone (in axial direction the volume extends far beyond the small bouton boundaries). This is less likely the case in FRAP experiments in the MFTs which measured 7 to 10 µm according to the authors: thus, rather volumes in the middle of the MFT at some distance to the AZs where the SV density is lowest (Figure 6) are probed. Although multiple or at least three measurements were made per terminal at different locations the large axial extent of the bleaching volume implies that mostly volumes away from the small AZs are covered. Given this and that the '3D reconstructions revealed the AZs of MFTs are small and surrounded by a cloud of vesicles that is highly variable in shape and extent', could the measured fast rates of fluorescence recovery predominantly reflect fast SV movements at the fringes of these SV clouds and even in-between clusters from different AZs? Here, overall SV densities are small and thus the dynamics might be much higher than within the densely packed areas close to the AZ, where SVs in EM tomograms even appear interconnected by short filaments, which might highly reduce their mobility.*

*In summary, given the experimental design of the FRAP measurements and the geometry of MFTs with their inhomogeneous SV distributions, there was concern about the claim that these results 'show diffusion ultimately limits vesicle supply during sustained high-frequency signaling at a central synapse'! SV supply and mobility at the AZ simply have not been and cannot be easily measured in this preparation.*

We agree that our measured long-time vesicle diffusion coefficient (D_long_) predominantly reflects the mobility of vesicles at the centre of the MFT rather than close to the AZ (as stated in the original version of the manuscript). Nevertheless, inspection of our reconstructions near the AZ shows that, except for a highly localized region extending only 100 nm from the AZ face, the vesicle density in the vesicle cloud is similar to that calculated within the non-mitochondrial space within the MFT interior (Figure 6, red symbols vs. dashed horizontal line). We now state this more explicitly in the Results (subsection “Quantification of the morphology around the AZ”).

It is important to stress that our approach is not to simply estimate D_long_ at the centre of the MFT and apply it to the AZ. Rather, we have developed an experimentally constrained quantitative model of vesicle mobility within crowded environments, which has enabled us to predict (and simulate) how vesicle diffusion changes as vesicles approach the AZ due to the increased vesicle density and presence of the membrane. This is possible because we account for the hydrodynamic and steric interactions arising from local variations in vesicle density, the cloud geometry and the presence of the AZ face itself. We now explicitly show in new Figure 7—figure supplement 1 how vesicle diffusion slows down in the vicinity of the AZ in our Monte Carlo simulations (down from 0.025 to 0.012 μm^2^/s, and even to 0.002 μm^2^/s in the presence of connectors).

Our model not only predicts the measured release rate during prolonged stimulation and the releasable pool at MFT AZs, it also predicts the diffusion coefficient of vesicles across different types of synapses. For example, our model predicts D_long_ of ribbon synapses in bipolar cells, which have a vesicle density comparable to MFTs (predicted 0.014 μm^2^/s vs. measured 0.015 μm^2^/s). It also predicts the lower mobility at the NMJ where vesicle density is more than double that within the MFT (0.33 vs. 0.17 volume fraction; predicted 0.006 μm^2^/s vs. measured 0.005 μm^2^/s). While measurements of vesicle mobility in small central synapses are more variable, our model predicts that D_long_ is low within the middle of the vesicle cluster in small boutons (prediction 0.002 μm^2^/s), which is similar to recent single-vesicle tracking measurements (0.003 μm^2^/s; Lee et al. 2012). Moreover, this predicted D_long_ falls within the wide range of estimates of vesicle mobility using FCS data (Figure 2, closed pink symbols). We have added a new table to the revised manuscript that compares the prediction of vesicle mobility across different synapses with existing measured values (Table 2). We have also modified the Discussion to include these new data (Comparison of vesicle mobility across synapses).

During the initial stages of sustained vesicular release, our results show that vesicle diffusion does not limit vesicle supply. This is because under resting conditions there is a high concentration of vesicles close to the AZ and supply rates via diffusion are high, even in the presence of tethers and connectors (1000 s^-1^; Figure 7) which reduce mobility to 0.002 μm^2^/s. At intermediate times (~100 ms), the vesicle density falls and vesicle supply is reduced to ~40 s^-1^ in the presence of tethers and connectors (Figure 7). At longer times, when the releasable pool (RP) becomes depleted, the vesicle concentration in the vicinity of the AZ drops well below that in the MFT interior (see new Figure 7—figure supplement 1). Under these conditions diffusion-based vesicle supply becomes rate limiting for release and the presence of connectors has little effect (Figure 7). We now link vesicle supply to depletion and diffusion more clearly in the revised manuscript (Results section, subsection “Models of diffusion-limited vesicle supply to the AZ during sustained release”), and explain why release becomes diffusion limited during sustained high-frequency signalling, even when diffusion is substantially slowed by connectors.

*2) An important conclusion of the paper is that "the geometry of the diffusible space determines the size of the RPV", based on the fact that back extrapolation of the cumulative release provides a similar pool size as measured in MFTs. However, before arriving at this conclusion several aspects should be addressed/discussed:*

*a) Using linear back extrapolation is known to underestimate the RRP due to the fact that the replenishment rate is not constant but depends on the depletion of the RRP (for a full RRP at steady-state the net replenishment rate is 0) (Neher, 2015). This error is larger for higher replenishment rates, which could explain the low RRP for the open geometry. Since in the simulations all vesicles are tractable, accurate estimates of the replenishment rates over time should be calculated and used for more accurate RRP estimates in the default- and open geometry.*

We agree with the referees that estimation of functional vesicle pools depend on a number of factors, but in this case it is possible that some confusion has arisen due to the nomenclature we used to describe the vesicle pools in the original submission. In the revised manuscript we now define 3 pools in the Introduction (first paragraph). Vesicles belonging to the readily releasable pool (RRP) are those that are docked and primed at the face of the AZ. Our EM data suggests there are ~2 vesicles docked at each small AZ (145 nm diameter). In contrast, vesicles in the releasable pool (RP) correspond to the ~300 non-docked vesicles per AZ, but they must be supplied to the AZ, then docked and primed (with all three steps referred to as reloading) before they transition to the RRP. Finally, vesicles in the reserve pool (R) correspond to a still larger pool that are supplied more slowly and are presumably further away from the AZ. We now illustrate these pools and their experimentally measured transition rates in a cartoon (Figure 8—figure supplement 1).

As the reviewers highlight, estimating the RRP can be difficult with back extrapolation because the pool size per connection is small (1-2 vesicles per AZ) and the reloading rate from the RP is fast at MFTs (40-80 s^-1^) and may be activity dependent (Hallermann et al. 2010). Moreover, the cumulative vesicle release plots of our Monte Carlo simulations did not reach steady state for control conditions until after 20 seconds. For this reason we did not attempt to estimate the RRP with the back extrapolation method. Instead, we estimated the RP (plus RRP) which is large (~300 vesicles per AZ), and since the R->RP replenishment rate is slow, the error due to back extrapolation should be relatively small. We now discuss the back extrapolation method and the pitfalls highlighted in Neher (2015) in the context of our analysis (first paragraph of Discussion).

For the open-geometry simulations there is a much higher diffusion-mediated supply rate (~55 Hz) than for the control AZ simulations. Since little vesicle depletion occurs during prolonged release under open-geometry conditions, equalization of release and replenishment rates occurred rapidly, resulting in a small number of vesicles being released before the release rate becomes constant. To examine this in more detail, we carried out more realistic simulations of release during a 100 Hz train, which included the measured RP->RRP reloading rate and a release probability (new Figure 8). While this did not alter the RP size estimated from the reconstructed AZ simulations with back extrapolation under control conditions (237 for instantaneous release vs. 217 for 100 Hz train), the RP estimated for open-geometry conditions was substantially reduced compared to instantaneous release (16 for instantaneous release vs. 2 for 100 Hz train) as predicted by Neher (2015). While it is clear that starting with a full RRP and having fast replenishment is likely to add some error in these latter estimates, the fact that the RP for the open geometry is 100-fold lower than the RP for the measured control cloud geometry, and is close to the size of the tiny RRP, confirms that the cloud geometry is a major determinant of the functional RP. Indeed, these simulations indicate that, when the diffusional barriers that define the extent of the vesicle cloud are removed, the RP no longer exists as an entity, with only the RRP and an effectively infinite reserve pool remaining. We now link these results to vesicle depletion and mobility during sustained release for the different simulation conditions (see new Figure 7—figure supplement 1). Moreover, since the physical extent of the pool identified with back extrapolation is unclear, we now describe it as the size of the pool of vesicles that can be depleted before release and supply rates equalize (excluding those replenished during that time; Results section, subsection “Models of diffusion-limited vesicle supply to the Az during sustained release”). This allows a more intuitive explanation of these effects since it is easy to understand why the geometry of the diffusible space around the AZ sets the R->RP supply rate.

*b) It is not clear how (molecular) priming is addressed in the model. Primed vesicles in the RRP are most likely docked at/tethered to the membrane via (partial?) assembly of SNARE-complexes and are therefore not free to diffuse (?). One would expect that this stabilizes an RRP of a certain size, independent of the geometry of the diffusible space but dependent on the number of available docking sites. This is supported by the fact that deletion of priming molecules like Munc13 and Munc18 produce a complete loss of RRP without affecting the geometry of the terminal. The impact of molecular priming on the RRP in the diffusion model should be discussed in the context of this evidence.*

We did not explicitly model molecular priming in our original simulations because it is fast in MFTs when compared with the R->RP replenishment rate. However, in light of the referees’ comments on estimating pool size, we have now included new simulations that mimic release during a 100 Hz train that include the delay introduced by docking and molecular/positional priming and the measured release probability at MFT AZs. Although these processes slowed vesicular release early in the train, they did not affect the limiting release rate at late times or the RRP pool size estimated via back extrapolation (new Figure 8).

Our EM data suggests there are ~2 vesicles next to the AZ face. If these correspond to docked and primed vesicles in the RRP, they are probably immobilized. However, since they represent less than 1% of the RP pool, it seems unlikely they can stabilize the 300 vesicles in the RP. Indeed, we have carried out an extensive set of simulations that include both tethers (filaments attaching vesicles to the AZ face) to multiple docking sites and connectors (filaments between vesicles), systematically varying the binding kinetics and number. While inclusion of these flexible 10 nm filaments within 150 nm of the release face can sustain the higher concentration of vesicles within 100 nm of the AZ release face, they still allow vesicle-AZ collisions to occur at well above the measured reloading rate (Figure 7). The fact that the concentration of vesicles beyond 100 nm of the release face is similar to that in the MFT interior is consistent with diffusional equilibration of the vesicle concentration and restricted diffusional space.

*c) A well-established method to assess the RRP is the application of hypertonic solutions. Interestingly, although this manipulation most likely has a strong impact on the geometry of the diffusible space (shrinkage), concentrations beyond 500mM have a strong effect on release kinetics but not on the total release or steady-state priming rates at the end of the stimulus. The authors should discuss how these findings can be reconciled with the model.*

To our knowledge, hypertonic solutions have not been used at the mossy fibre to granule cell synapse. A recent paper (Schotten et al. *eLife*, 2015) argues that their effects are predominantly to lower the activation energy for fusion rather than the AZ hit rate. This conclusion is based on measurements of hypertonic-solution evoked release and modelling of the activation energy for vesicle fusion and on the observation that hypotonic solutions specifically release primed vesicles (Rosenmund and Stevens, 1996), which are bound to the plasma membrane with the fusion machinery preassembled, and are not therefore affected by diffusion-limited vesicle supply. Indeed it is difficult to predict how rapid cell shrinking would affect diffusion-based supply, since it is unclear whether the increase in the concentrations of vesicles or slowing of diffusion by steric and hydrodynamic interactions and viscosity will increase or decrease the AZ hit rate. Since our study is focused on supply from the RP and reserve pools, we feel that discussing how sucrose affects the RRP is beyond the scope of this study.

*3) Priming/RRP replenishment is sped up in a Ca^2+^ dependent manner during high frequency stimulation (Sakaba & Neher, 2001). This suggests an active flow of vesicles in the direction of the AZ during or a change in mobility properties of the vesicles. The authors should discuss how this fits in their model.*

Our results suggest that diffusion-mediated vesicle supply to the AZ is fast at early times and that most of the RP->RRP vesicle reloading time (12-20 ms) is taken up by docking and priming. This suggests that, at the onset of high-frequency signalling, Ca^2+^-dependent processes that speed docking and priming would be most effective, but that increasing vesicle mobility or reducing the immobile fraction would be effective once vesicle depletion occurs. Interestingly, our simulations reveal that when vesicle depletion lowers vesicle density at the AZ, vesicle mobility is automatically increased due to reduced steric and hydrodynamic interactions (see new Figure 7—figure supplement 1). We now discuss our findings in the context of activity-dependent increases in reloading in the Discussion (Discussion section, subsection “Supply of vesicles to the AZ and structural correlates of the reasonable pool”).

*4) Software or scripts should be made available to the community as supplementary material via eLife or via a public repository for computational models for neuroscience.*

We will make the simulators and scripts available for download from GitHub.